# CURRICULUM METRIC LEARNING FOR ROBUST IMAGE RETRIEVAL

## ABSTRACT

Deep Metric Learning (DML) is integral to retrieval systems, aiming to find relevant items for a query sample. While effective, DML models are susceptible to adversarial ranking attacks, where adversarial perturbations alter retrieval rankings. This work challenges the conventional initialization of DML defenses with pretrained ImageNet weights, hypothesizing that this approach is suboptimal. Learning approaches for robust retrieval representations must accomplish two goals: (1) learn semantically meaningful representations, and (2) learn robust representations resistant to ranking attacks. We propose a curriculum learning strategy for robust retrieval, which involves a two-step learning process: (1) acquiring robust features resistant to adversarial perturbations, and (2) applying robust metric learning for precise retrieval. Our approach starts with adversarial pre-training, positing that models trained to withstand adversarial perturbations ($\Delta$) offer a stronger foundation for robust retrieval. We quantitatively demonstrate that robust feature representations, acquired through adversarial training, inherently exhibit resilience against ranking attacks. This adds resistance to adversarial ranking attacks as an additional benefit of robust representations, in addition to for example better transferability. By leveraging these robust features as a foundation followed robust metric learning, we significantly enhance the recall of benign examples and fortify defenses against adversarial ranking attacks, achieving state-of-the-art results across three benchmark datasets (CUB, CARS, SOP).

## 1 INTRODUCTION

The goal of deep metric learning (DML) is to learn a representation or embedding space for data such that similar samples are closer to each other while dissimilar samples are farther apart. In doing so, metric learning encodes semantically relevant distances between embeddings in high dimensional space. Deep metric learning is applied to many tasks - e.g. image retrieval (Gordo et al., 2017), face recognition (Hu et al., 2014; Wang et al., 2018), person re-identification (Yi et al., 2014), and object tracking (Hu et al., 2016) - and is used in many state-of-the-art self-supervised learning approaches (Chen et al., 2020; Chen & He, 2021). DML facilitates comparisons and retrieval of similar samples, despite variations in lighting, pose, or background. The ultimate objective is to enable efficient and effective similarity-based reasoning and search in high-dimensional data spaces. Metric learning

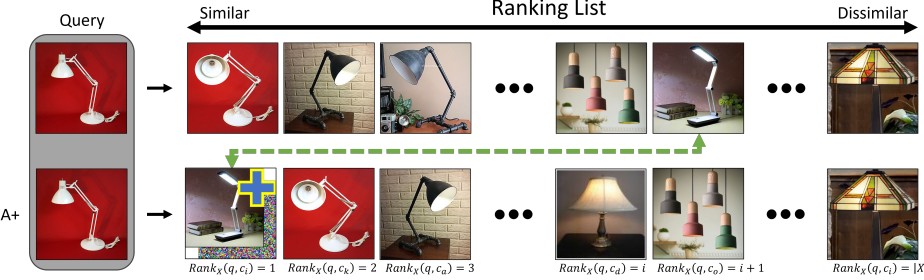

Figure 1: Adversarial attacks against DML models alter the rank-ordering of a retrieved list. A candidate attack is applied to the image of the ornate lamp to artificially promote it in the retrieval search.

techniques typically leverage deep neural networks to learn powerful representations that capture relevant characteristics and semantics of the data, facilitating accurate similarity comparisons and supporting various applications. However, metric learning approaches that leverage deep neural networks are vulnerable to adversarial examples (Szegedy et al., 2013) raising various concerns in applications such as e-commerce (adversarial item promotion) (Liu & Larson, 2021) or face verification. Furthermore, recent works (Tolias et al., 2019; Zhou et al., 2020a) developed various attacks methods that directly perturb the rank-order of examples in a retrieved list. In Figure 1 we see an example of a Candidate Attack (CA) (Zhou et al., 2020a), which raises the rank of a single candidate (database) image with respect to a set of Queries, is provided.

**How can we improve robust representation learning?** Pre-training techniques can be used to imbue deep models with priors from large-scale datasets (Russakovsky et al., 2015) or related tasks (Zamir et al., 2018). By pre-training a model on a large corpus of data, such as ImageNet (Russakovsky et al., 2015) for computer vision or a massive text corpus for natural language processing (Touvron et al., 2023), a model can learn general priors about the world that are applicable to a wide range of downstream tasks. This pre-training enables models to capture important features and regularities that are beneficial for subsequent fine-tuning on specific tasks, leading to improved generalization and efficiency (Kornblith et al., 2019).

Recent methods for improving models' robustness to adversarial ranking attacks (Zhou et al., 2020a; 2021) are inspired by adversarial training of Madry et al. (2018), though they replace the inner maximization problem with easier tasks to avoid model collapse Zhou & Patel (2022). Deep Metric Learning, like reinforcement learning, is notoriously tricky to "get right". Successful learning requires proper initialization (Andrychowicz et al., 2021), training procedure (Gordo et al., 2017), or sampling strategy (Wu et al., 2017). The goal of adversarial ranking defenses is to learn representations that are simultaneously (1) robust to ranking attacks, and (2) learn semantically meaningful representations. In contrast to previous adversarial ranking defenses (Zhou et al., 2021; Zhou & Patel, 2022), we take inspiration from curriculum learning (Bengio et al., 2009) and propose to accomplish the task of learning representations that are simultaneously robust and meaningful by breaking down the learning into two sub-tasks: (1) learning robust features, and (2) learning robust semantic representations starting from robust features. In our case, robust features come from models subjected adversarial training procedure of Madry et al. (2018) on the ImageNet dataset (Russakovsky et al., 2015) for different of $\epsilon$ (maximum allowable perturbations for adversarial examples that are used to train the model).

**Should adversarial robustness help robust metric learning?** It is a priori unclear if transferring robust representations from disparate tasks could help robust metric learning. However, as robust models are trained to be invariant to a set of perturbations $\delta \in \Delta$ (Engstrom et al., 2019), we hypothesize such representations be more invariant to ranking attacks and to be a better starting point for robust ranking defenses. Figure 2 (right) shows the performance of a DML model when initialized with robust weights obtained from an adversarially trained ImageNet model (Madry et al., 2018). As a result, the ranking method develops a robustness to ranking-based attacks as well. While the added robustness comes at a cost of decreased performance for benign examples, weak adversarial defenses such as "Embedding-shifted Triplet" become quite formidable when initialized with a robust prior. This improvement is measured via the Empirical Robustness Score (Zhou et al., 2021)) and can be observed and Figure 2 (middle).

**Our contributions** We are the first, to the best of our knowledge, to explore and quantify the adversarial ranking robustness of adversarially trained models. We show that robust models trained with adversarial training (Madry et al., 2018) are, to various degrees, already resistant to recently proposed ranking attacks (Tolias et al., 2019; Zhou et al., 2020a; Feng et al., 2020). This opens a new avenue in understanding how adversarial training affects learned features, and adds ranking robustness as an additional benefit of robust representations, in addition to for example better transferability (Salman et al., 2020), or explainability Bansal et al. (2020). Our results directly support hypotheses suggesting that adversarial robustness leads to improved feature representations (Engstrom et al., 2020; Allen-Zhu & Li, 2022; Salman et al., 2020). Furthermore, we propose a curriculum metric learning approach which decomposes the task of learning robust retrieval representations into two progressively harder task. First, the deep networks models learn robust features via adversarial training (easier task) by optimizing a robust classification objective on a large-scale dataset, followed

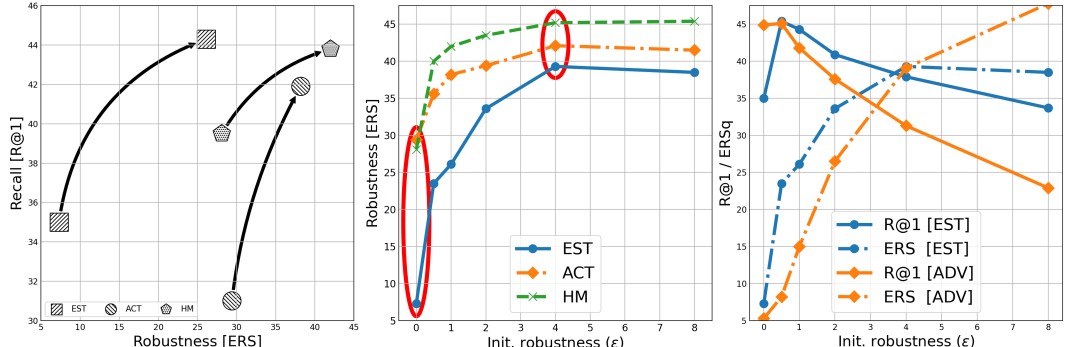

Figure 2: Comparison of model performance on benign and adversarial examples on the CUB-200-2011 dataset (Welinder et al., 2010). *Left* figure shows that robust initialization significantly improves both recall and ranking robustness for the state-of-the-art DML defenses: a) Embedding-Shifted Triplet (EST) , (b) Anti-Collapse Triplet (ACT) , and (c) Hardness Manipulations (HM) . *Middle* figure demonstrates that EST, while initially weak become quite a formidable defense with robust initialization. Lastly, the figure on the *right* shows that $l_\infty$ robust ImageNet models are to various degrees robust to ranking attacks (ERS[ADV]) (though their R@1 suffers). Further we see that if the EST defense is applied on top of robust features, its performance significantly improves both in terms of recall and ranking robustness.

by robust metric learning which aims to enforce semantic representations for precise retrieval. We show robust features critical factor in improving robust metric learning for retrieval, significantly improving the effectiveness of adversarial ranking defense methods Zhou et al. (2020a; 2021); Zhou & Patel (2022). We establish such trends on three common benchmark datasets: CUB-200 (Welinder et al., 2010), CARS-196 (Krause et al., 2013), and Stanford Online Products (SOP) (Oh Song et al., 2016).

## 2 RELATED WORK

**Adversarial robustness.** Though neural networks have been in the forefront of state-of-the-art research in many branches of computer vision, they exhibit some weaknesses, among them vulnerability to adversarial examples (Szegedy et al., 2013; Goodfellow et al., 2015), which completely change the output of the model. Recent years have seen many attempts to propose defenses against such examples, e.g. stochastic activation pruning (Dhillon et al., 2018), input pre-processing via transformations (e.g. JPEG compression) (Guo et al., 2018), or randomization (input re-scaling) (Xie et al., 2018). However, as Athalye et al. (2018) show, many of these defenses can be easily circumvented. One of the most effective approaches to defending against adversarial attacks is shown to be adversarial training (Madry et al., 2018; Athalye et al., 2018). **Deep Metric Learning.** Deep metric learning can be applied to a wide range of problems where measuring and understanding similarity or dissimilarity between data points is crucial, including image retrieval (Gordo et al., 2017), semantic segmentation (Cen et al., 2021), or product recommendation (Campo et al., 2018). Though DML approaches can be powerful, they require much care in implementing properly - proper weight initialization, training procedure (Gordo et al., 2017), sampling strategy (Wu et al., 2017), and choice of proper learning loss (e.g. contrastive (Radenović et al., 2019), triplet (D'Innocente et al., 2021), or mAP (Revaud et al., 2019) loss). **Adversarial ranking attacks and defenses** In the recent years, many adversarial ranking attacks which subvert the retrieval results have been proposed(Tolias et al., 2019; Wang et al., 2020; Feng et al., 2020; Zhou et al., 2021). Despite the abundance of efforts to subvert the retrieved ranking, only a few efforts focus on proposing defenses against adversarial ranking attacks, namely "Embedding-Shifted Triplet" defense (Zhou et al., 2020a), "Anti-Collapse" Triplet defense (ACT)(Zhou et al., 2021) and "Hardness Manipulation" (HM)(Zhou & Patel, 2022). **Curriculum learning (CL)** is a framework of machine learning training strategies where tasks or data samples are sequenced into a progressively more difficult curriculum in order to solve problems that might be too hard to solve from scratch (Bengio et al., 2009; Narvekar et al., 2020). CL is a popular framework in many applications, such as reinforcement learning (Narvekar et al., 2020), computer vision (Wang et al., 2021). Previous works that touched on robust metric learning (Zhou et al., 2020b; Castells et al., 2020) focus on the notion of robustness against noisy samples, and thus significantly differ from our application of defending against adversarial ranking attacks.

## 3 BACKGROUND

**Adversarial ranking attacks against DML models.** In retrieval systems, adversarial perturbations are added to either a query image or a candidate (retrieved) image where the goal is to change the rank ordering of the retrieved list. For an example of an adversarial ranking attack, see Figure 1. In this case, an adversarial perturbation is applied to a candidate in the retrieved list (top of Figure 1) such that it raises its rank for several query images (bottom of Figure 1). Alternatively, attackers may perturb the query $q$ to force the embedding distance between itself and items that nominally match it to be larger, thus moving the candidate matches down the retrieved list.

**Defending against robust adversarial ranking attacks.** Current ranking defenses (EST, ACT, HM)[1] can be viewed as relaxations of adversarial training Madry et al. (2018). Similar to other metric learning approaches, these ranking defenses minimize the triplet ranking loss, however propose different ways of generating adversarial images in the triplets to guide the learning process. In the **EST** defense, the goal is to find adversarial examples that shift the embedding vector as far away from the original vector as possible and optimize a defensive triplet loss, where the clean examples are replaced by their adversarial counterparts,

$$L_{\text{EST}} = L_{\text{T}}(\tilde{a}, \tilde{p}, \tilde{n}; \gamma) \tag{1}$$

where $L_{\text{T}}$ is a standard triplet loss, $\gamma$ is the margin parameter, $\tilde{a} = \phi(A + r^*)$, and $r^* = \arg\max_r d_\phi(A + r, A)$. The $\tilde{p}$ and $\tilde{n}$ are obtained similarly. Please refer to the Appendix A.7 for a more detailed description on how to find adversarial examples for the EST defense.

**Learning robust retrieval representation using a sub-task curriculum.** Curriculum learning framework mimics the way humans learn, starting with simpler concepts and gradually moving to more complex ones (Narvekar et al., 2020). Over time, as the tasks become more challenging, the model continues to refine and develop its capabilities, leading to more effective learning and better performance on complex tasks.

Learning robust retrieval representations aims to achieve goals: (a) learn semantically meaningful representations, and (b) and learn robust representations. Trying to achieve both goals can lead to competing objective at the beginning, and therefore we propose to make the learning process easier by breaking it down into two progressively harder sub-tasks. The first, and easier, sub-task will involve learning robust feature representations through adversarial pre-training. The second and final sub-task will aim to learn semantically meaningful features (using metric learning) while trying to preserve or improve further their robustness (by using adversarial versions of input samples).

**Robust optimization as a feature prior.** In our work, following Engstrom et al. (2019); Ilyas et al. (2019); Salman et al. (2020), we enforce the robustness prior on the initial weights by forcing the network to solve the robust optimization objective Athalye et al. (2018) for ImageNet classification,

$$\theta^* = \arg\min_\theta \mathbb{E}_{(x,y)\sim\mathcal{D}} [\mathcal{L}(x,y)] \implies \theta^* = \arg\min_\theta \mathbb{E}_{(x,y)\sim\mathcal{D}} \left[ \max_{\delta\in\Delta} \mathcal{L}(x+\delta, y) \right]. \tag{2}$$

Note that as models minimize the robust objective of Eq. 2, they are expected to be invariant to a set of perturbations $\delta \in \Delta$ (Engstrom et al., 2019); and as such, we can expect these representations be more invariant to ranking attacks and to be a better starting point for robust ranking defenses.

## 4 CURRICULUM METRIC LEARNING FOR ROBUST IMAGE RETRIEVAL

**Experiments** We perform experiments on the following three datasets: CUB-200-2011 (CUB) (Welinder et al., 2010), Cars-196 (CARS) (Krause et al., 2013), and Stanford Online Products (SOP) (Oh Song et al., 2016). For easy of reproducibility, our experiments are build on top of the RobRank GitHub[2] repository, which implements the state-of-the-art deep ranking defenses: (a) Embedding-Shifted Triplet (EST) (Zhou et al., 2020a), (b) Anti-Collapse Triplet (ACT) (Zhou et al., 2021), and (c)

---

[1]a) Embedding-Shifted Triplet (EST) (Zhou et al., 2020a), (b) Anti-Collapse Triplet (ACT) (Zhou et al., 2021), and (c) Hardness Manipulations (HM)(Zhou & Patel, 2022)

[2]https://github.com/cdluminate/robrank

| Dataset | Defense | $\epsilon$ | Benign Example | | | | White-Box Attacks for Robustness Evaluation | | | | | | | | | | ERS↑ |
| | | | R@1↑ | R@2↑ | mAP↑ | NMI↑ | CA+↑ | CA-↓ | QA+↑ | QA-↓ | TMA↓ | ES:D↓ | ES:R↑ | LTM↑ | GTM↑ | GTT↑ | |
|---|---|---|---|---|---|---|---|---|---|---|---|---|---|---|---|---|---|
| CUB | AT | 0.0 | 44.9 | 58.6 | 16.2 | 55.1 | 0.0 | 100.0 | 0.0 | 99.7 | 0.9 | 1.3 | 0.4 | 0.0 | 12.0 | 0.0 | 5.3 |
| | AT | 0.5 | 45.1 | 57.8 | 16.1 | 52.6 | 0.4 | 99.2 | 0.2 | 95.5 | 0.9 | 1.0 | 3.1 | 0.7 | 12.4 | 0.0 | 8.2 |
| | AT | 1.0 | 41.8 | 54.5 | 14.4 | 49.2 | 1.9 | 91.1 | 1.7 | 77.0 | 0.8 | 0.8 | 10.8 | 7.5 | 17.0 | 0.1 | 15.0 |
| | AT | 2.0 | 37.6 | 49.5 | 12.4 | 46.8 | 6.7 | 70.7 | 8.0 | 43.3 | 0.8 | 0.6 | 18.1 | 18.8 | 20.5 | 1.4 | 26.5 |
| | AT | 4.0 | 31.3 | 42.0 | 10.1 | 42.6 | 15.2 | 39.6 | 19.9 | 15.8 | 0.8 | 0.4 | 20.2 | 23.7 | 20.6 | 7.1 | 39.1 |
| | AT | 8.0 | 22.9 | 32.1 | 7.2 | 37.4 | 24.5 | 18.5 | 29.5 | 6.2 | 0.7 | 0.3 | 19.6 | 21.0 | 17.6 | 18.4 | 47.8 |

Table 1: Comparison of model performance of $l_\infty$ robust ImageNet models for various robustness values $\epsilon$ on benign and adversarial examples on the CUB-200-2011 dataset(note $\epsilon = 0$ denotes a non-robust initialization). Performance on benign examples is summarized by recall (R@1) in the second column. Ranking robustness is tested across an ensemble of adversarial ranking attacks, e.g. Candidate Attack (CA +/-), and summarized by the Empirical Robustness Score (ERS) in the last column. See Appendix A.9 for description of adversarial ranking attacks used to obtain ERS. The "↑" mark means "the higher the better", while "↓" means the opposite.

Hardness Manipulations (HM) (Zhou & Patel, 2022). In the following paragraphs we briefly outline the training and evaluations protocol for our models. For further details, please see the *Appendix* at the end of the paper.

**Evaluation protocol** To evaluate the impact of robust features on deep ranking defenses, we assess six sets of weights characterized by adversarial perturbations on ImageNet: $\epsilon \in \{\frac{0}{255}, \frac{0.5}{255}, \frac{1}{255}, \frac{2}{255}, \frac{4}{255}, \frac{8}{255}\}$. The baseline model is initialized with standard ImageNet weights (trained only on clean images), and is denoted as $\epsilon = \frac{0}{255}$ for consistency in labeling. Robust ImageNet weights are obtained from models trained using adversarial training (Madry et al., 2018) on adversarial images with maximum allowable $l_\infty$ perturbations of $\epsilon \in \{\frac{0.5}{255}, \frac{1}{255}, \frac{2}{255}, \frac{4}{255}, \frac{8}{255}\}$. Following recent work on adversarial ranking defenses (Roth et al., 2020; Zhou & Patel, 2022), we benchmark the performance of each model on its ability to perform image retrieval on unperturbed samples and report Recall@1 (R@1), Recall@2 (R@2), mean Average Precision (mAP), and Normalized Mutual Information (NMI). **Empirical Robustness Score** (Zhou et al., 2021) is a normalized score (0-100) that summarizes a model's robustness against an ensemble of adversarial ranking attacks. Model adversarial ranking robustness (ARR) against ranking attacks is quantified by "Empirical Robustness Score" (ERS) (Zhou et al., 2021). Similar to Auto-attack metric (Croce & Hein, 2020) for classification, to obtain the ERS, one performs an ensemble of adversarial ranking attacks (to perturb the retrieved list for a given query) and computes the average success rate across the ensemble of ranking attacks. For a more detailed description of the adversarial ranking attacks used to obtain ERS to test ARR, please see the Appendix A.9. Similar to Zhou & Patel (2022), each of the adversarial attacks is optimized using $\eta = 32$ PGD iterations to ensure the strongest possible adversary.

### 4.1 ROBUST FEATURES ARE ALREADY RESISTANT TO ADVERSARIAL RANKING ATTACKS

We first explore the ranking robustness of robust features of models tasked to optimize the robust classification loss of Equation 2 on the ImageNet dataset. As previously stated, we hypothesize that such models are achieve higher ranking robustness (RR) than a cleanly trained model. Table 1 shows the performance metrics for retrieval on clean examples (R@1) and ranking robustness (ERS) of adversarially trained models. To compute the features for retrieval of the classification models, we simply cut off the topmost classification layer and normalize the penultimate layer features. As we can see from the Table 1, training the classification models to be more robust, i.e. allowing for higher maximum perturbation $\epsilon$ leads to the higher is their ranking robustness (ERS goes up). Though we also see a parallel to the accuracy-robustness trade-off (Tsipras et al., 2019), where higher robustness is met with a decrease in recall on clean examples.

### 4.2 ROBUST FEATURES IMPROVE CONVERGENCE OF ROBUST DML

This section explores the effect of adversarial pre-training on the convergence of robust deep metric learning. First we show how robust features drastically improve the learning during EST defense. In Table 1 (left), we can see that increasing the robustness $\epsilon$ of robust features (i.e. robust initialization) significantly improves its ranking robustness (ERS) of the EST defense. Figure 3 shows the evolution

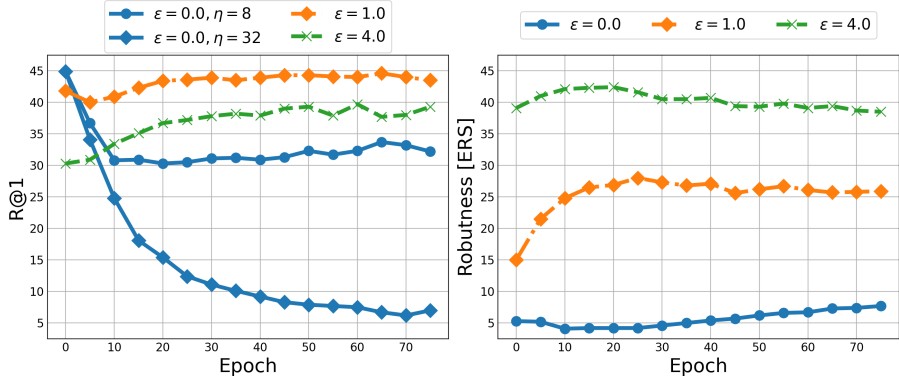

Figure 3: Figures showing the impact of robust initialization on model convergence. *Left* and *right* figures show the evolution of recall and robustness (ERS) respectively from different initializations as it progresses through training. $\epsilon$ denotes the pre-training robustness magnitude and $\eta$ denote indicates the number of steps used to generate adversarial triplets.

of Recall (R@1) (left) and robustness (right) during training when the EST defense starts from robust vs. non-robust features ($\epsilon = 0$). As we can see, if we apply the EST defense to non-robust features ($\epsilon = 0$, the model's recall progressively decreases from $R@1 \approx 45$ at the beginning of training and gets progressively worse, while we see no significant improvement in robustness. While starting the EST defense from robust features allows the model to achieve good recall and improve its robustness (see $\epsilon = 1.0$ or $\epsilon = 4.0$ on Figure 3).

Robust features can also improve various aspects of the Hardness Manipulations (HM) defense. If initialized from non-robust features, several combinations of triplet sampling strategies for the HM defense result in model collapse Zhou & Patel (2022). Robust features goes a long way to stabilizing and improving HM training improving both recall and ranking robustness, as only 2 of out the 25 possible combinations resulted in significant drop in clean recall, while almost half of combinations collapse when HM is initialized with non-robust features. Please see Table A1 for tabulated results of clean recall and ranking robustness for various triplet sampling strategies. In further experiments, for the HM defense we use random and soft hard sampling for source hardness sampling, while using linear gradual adversary (LGA) for destination hardness (Zhou & Patel, 2022) as these combinations achieve both high recall and high ranking robustness (See Table A2).

### 4.3 ROBUST FEATURES IMPROVE RECALL AND RANKING ROBUSTNESS OF DML DEFENSES

This section will explore the effect on robust features on existing DML defenses: Embedding-Shifted Triplet (EST), Anti-collapse triplet (ACT) and Hardness Manipulations (HM). To understand how each of the DML defenses is affected by robust features, we initialize the ResNet-18 DML model with an $l_\infty$ robust ImageNet checkpoint for various values of $\epsilon \in \{0.5/255, 1/255, 2/255, 4/255, 8/255\}$. Each defense uses $\eta = 8$ PGD steps to generate adversarial images for metric learning, and the random ($\mathcal{R}$) and soft-hard ($\mathcal{S}$) sampling strategy for triplets. We can see the positive impact of robust features benign (R@1) and robust (ERS) examples in Figure 4.

Though the EST defense was previously handily beat by both ACT and HM (Zhou & Patel, 2022), robust features provide significant improvement in model recall and ranking robustness. EST[$\mathcal{R}$] jumps in performance from 35.0 R@1 / 7.3 ERS when $\epsilon = 0$ to 37.9 R@1 / 39.3 ERS when $\epsilon = 4$, which represents more than a 500% improvement in ranking robustness (ERS) with proper initialization while making it significantly more competitive with the other defenses. Interestingly, soft-hard sampling is initially better for EST on both benign and adversarial examples, whereas ERS improves on EST[$\mathcal{R}$] over EST[$\mathcal{S}$] for all values of $\epsilon > 0$. Soft-hard sampling does outperform random sampling in terms of R@1. Interestingly, for EST[$\mathcal{R}$] the ERS scores peaks at $\epsilon = 4$ and then drops, whereas for EST[$\mathcal{S}$], ERS keeps improving with increasing initialization robustness.

The Anti-Collapse Triplet (ACT) defense (ACT[$\mathcal{R}$] and ACT[$\mathcal{S}$]) shows very similar trend to ESR[·] defense. Robust features in general improves performance on both benign and adversarial examples, random sampling outperforms soft-hard sampling, and ERS peaks at $\epsilon = 4$ for random sampling. For

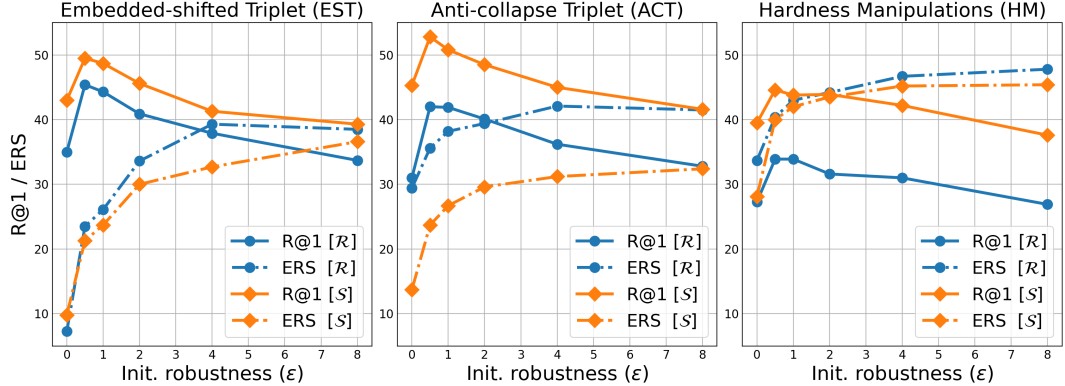

(a) Understanding the effect sampling strategy on the ranking defense methods (EST (left), ACT (middle), and HM (right)) with varying initialization robustness.

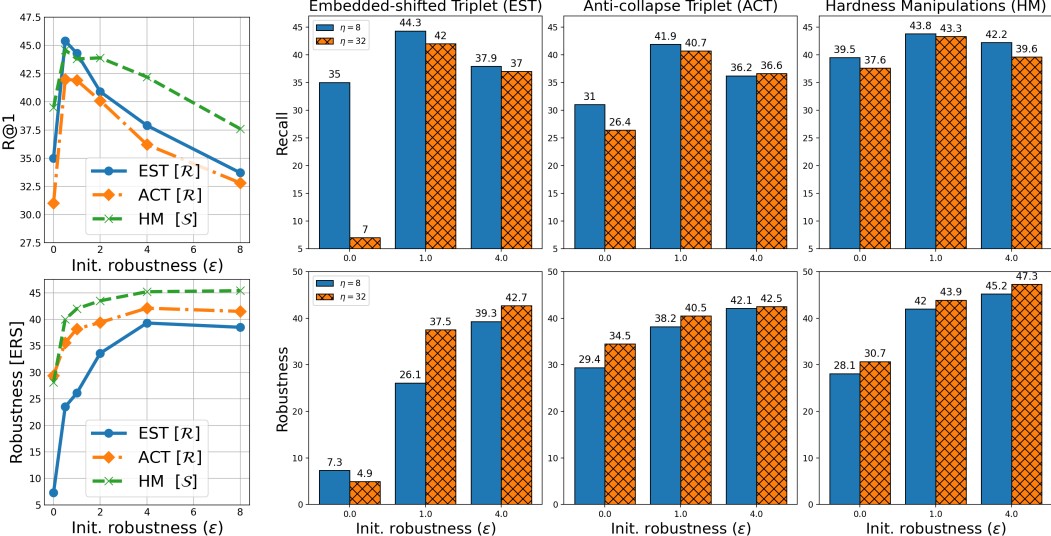

(b) Comparison of best performing versions (sampling) of various defenses.

(c) Effect of $\eta$ and $\epsilon$ on state-of-the-art defenses: (left) Embedding-Shifted Triplet (EST), (middle) Anti-Collapse Triplet (ACT), and (right) Hardness Manipulations (HM).

Figure 4: Understanding the effect of robust features (initialization robustness) on DML defense methods.

random sampling, ACT[$\mathcal{R}$] outperforms EST[$\mathcal{R}$] when initialized with robust checkpoint (e.g. $\epsilon = 4$, 36.2 R@1 / 42.1 ERS (ACT) vs 37.9 / 39.3 (EST)), but by a smaller margin than initializing the defenses with regularly trained ImageNet models - 31.0 R@1 / 29.4 ERS (ACT) vs 35.0 / 7.3 (EST). Surprisingly, EST is more robust as compared to ACT for larger value of $\epsilon = 0$ when Soft-hard [$\mathcal{S}$] triplet sampling strategy is employed.

Lastly, we consider Harness Manipulation defense. In this case, we see that robust initialization significantly improves performance on both benign and adversarial examples. With $\epsilon = 0$, HM[$\mathcal{R}, g_{\mathsf{LGA}}$] achieved better ERS and lower R@1 as compared to HM[$\mathcal{S}, g_{\mathsf{LGA}}$]. However for $\epsilon > 0$, HM[$\mathcal{S}, g_{\mathsf{LGA}}$] achieves a much better R@1 and comparable ERS making it the superior strategy for HM defense.

**Effect of $\eta$ and $\epsilon$ on ranking defenses** Next we explore how does model performance scale with increasing of the number of PGD iteration steps for finding adversarial triplets in the various DML defenses EST[$\mathcal{R}$], ACT[$\mathcal{R}$], and HM[$\mathcal{S}, g_{\mathsf{LGA}}$]. We see in Figure 4c that for all values of $\eta$, robust initialization improves both R@1 and ERS for all methods. Surprisingly, when the EST defense is initialized with clean weights, increasing number of PGD iterations $\eta$ leads to a significant drop in performance on both benign and adversarial examples. However a large improvement in R@1 and ERS can be seen when robust weights are used for initialization and even further improvement in

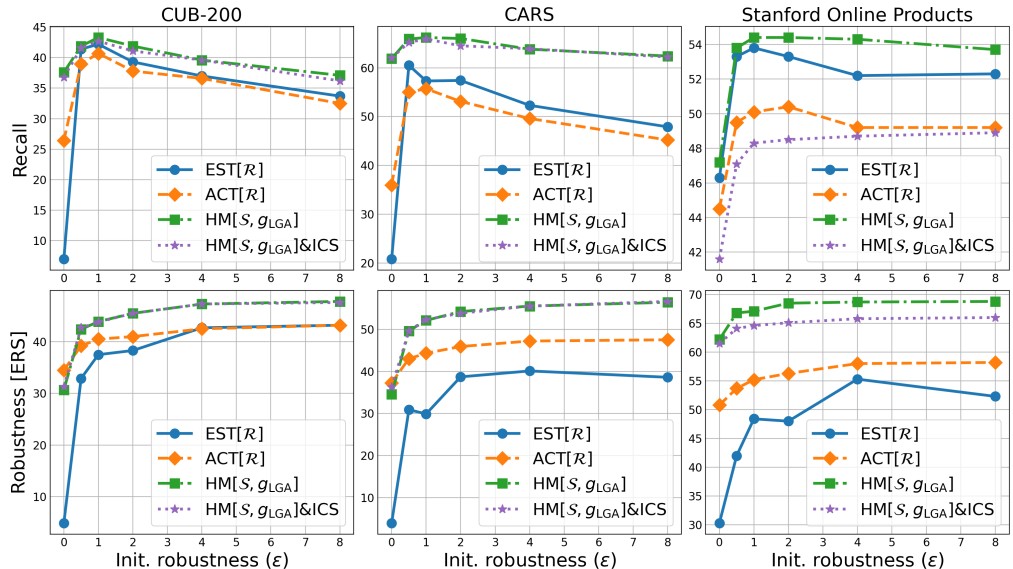

Figure 5: Comparison of recall (R@1 on benign examples) and robustness (ERS) of the various SOTA defense as a function of robust model initialization across three benchmarks datasets: (left) CUB (Welinder et al., 2010), (middle) CARS (Krause et al., 2013), and (right) SOP (Oh Song et al., 2016).

ERS with increasing number of PGD steps (which is expected as larger number of PGD iterations allows us to find harder adversarial examples and thus should lead to larger robustness).

## 4.4 COMPARISON OF CURRICULUM METRIC LEARNING TO SOTA

Lastly, we explore the effect of robust initialization of robust defenses across the three benchmark datasets: CUB, CARS, and SOP. Table 2 and Figure 5 summarize the results for the three datasets and various defense methods. In Table 2, defenses initialized with robust features that follow are curriculum learning approach are denoted as CML-*defense*, e.g. CML-EST[$\mathcal{R}$].

We can see that robust initialization leads to a significant improvement for EST[$\mathcal{R}$] on both benign and adversarial examples. From the bar plots in 4c, we see that ERS[$\mathcal{R}$] suffers and catastrophic drop in R@1 when number of PGD iterations was increased from 8 to 32, and achieved much lower robustness as compared to other defense methods when initialized with non-robust features. However, robust initialization before EST[$\mathcal{R}$] leads to a significant improvements in both R@1 and ERS for all values of $\eta$ across each dataset - in fact robustly initialized EST[$\mathcal{R}$] beats even the best defense method (HM) if it is applied to non-robust features ($\epsilon = 0$ and $\eta = 32$).

Figure 5 shows a visual comparison of model performance (R@1) for difference defense methods across benchmark datasets. In general, robust initialization of defenses leads to a considerable increase in both R@1 and ERS. More specifically, robust initialization ($\epsilon = 4/255$) leads to 51.1%, 52.8%, and 10.5% improvement in robustness in terms of ERS for CUB, CARS, and SOP datasets respectively. From the Figure, we can see that the ICS loss term can be helpful in increasing model performance on the CUB and CARS datasets, however it hurts the performance on the SOP dataset, dropping R@1 / ERS from 54.7 / 68.7 to 48.7 / 65.8 when model is initialized with $l_\infty$ $\epsilon = \frac{4}{255}$ robust weights.

## 4.5 WHY DOES ROBUST PRE-TRAINING IMPROVE ROBUST METRIC LEARNING?

We hypothesize that robust pre-training crucial for helping shape the metric space during training to allow for easier optimization of the robust triplet loss. In the case of EST, we can see from Figure 3 that as training proceeds, the recall slowly collapses - which we think is a result of the network trying to optimize two competing objectives - maximizing robustness while improving recall. At larger values of $\epsilon$, we see the training progress as hypothesized in our CML description in Section 3, i.e.

| Dataset | Defense | $\eta$ | Benign Example | | | | White-Box Attacks for Robustness Evaluation | | | | | | | | | | ERS↑ |
|---|---|---|---|---|---|---|---|---|---|---|---|---|---|---|---|---|---|
| | | | R@1↑ | R@2↑ | mAP↑ | NMI↑ | CA+↑ | CA-↓ | QA+↑ | QA-↓ | TMA↓ | ES:D↓ | ES:R↑ | LTM↑ | GTM↑ | GTT↑ | |
| CUB | N/A | N/A | 53.9 | 66.4 | 26.1 | 59.5 | 0.0 | 100.0 | 0.0 | 99.9 | 0.883 | 1.762 | 0.0 | 0.0 | 14.1 | 0.0 | 3.8 |
| | HM[$\mathcal{S}, g_{\text{LGA}}$]&ICS ($\epsilon = 0$) | 8 | 38.1 | 48.9 | 22.7 | 49.8 | 11.3 | 47.8 | 11.8 | 43.2 | 0.8 | 0.5 | 14.7 | 12.3 | 18.2 | 0.9 | 29.1 |
| | CML-EST[$\mathcal{R}$] | 8 | 37.9 | 49.6 | 22.5 | 50.4 | 16.1 | 36.7 | 17.0 | 25.8 | 0.5 | 0.6 | 25.2 | 22.3 | 20.9 | 3.6 | 39.3 |
| | CML-ACT[$\mathcal{R}$] | 8 | 36.2 | 47.6 | 22.6 | 49.7 | 20.6 | 30.0 | 21.3 | 22.1 | 0.4 | 0.7 | 19.4 | 20.6 | 21.5 | 4.2 | 42.1 |
| | CML-HM[$\mathcal{S}, g_{\text{LGA}}$] | 8 | 42.2 | 53.1 | 25.1 | 51.6 | 20.8 | 25.9 | 21.8 | 18.3 | 0.5 | 0.5 | 27.6 | 28.2 | 25.5 | 8.0 | 45.2 |
| | CML-HM[$\mathcal{S}, g_{\text{LGA}}$]&ICS | 8 | 41.1 | 51.9 | 25.1 | 51.8 | 21.8 | 21.3 | 25.5 | 14.9 | 0.8 | 0.3 | 29.0 | 30.0 | 26.1 | 10.4 | 46.0 |
| | HM[$\mathcal{S}, g_{\text{LGA}}$]&ICS($\epsilon = 0$) | 8 | 36.8 | 47.8 | 21.3 | 48.7 | 13.4 | 44.9 | 13.9 | 38.8 | 0.8 | 0.6 | 15.1 | 13.0 | 18.7 | 1.3 | 31.3 |
| | CML-EST[$\mathcal{R}$] | 32 | 37.0 | 49.0 | 21.0 | 49.6 | 19.8 | 36.6 | 22.3 | 18.2 | 0.5 | 0.6 | 26.9 | 25.3 | 21.6 | 4.4 | 42.7 |
| | CML-ACT[$\mathcal{R}$] | 32 | 36.6 | 46.5 | 21.6 | 48.9 | 21.1 | 28.4 | 21.6 | 21.3 | 0.4 | 0.7 | 19.8 | 19.3 | 21.3 | 3.9 | 42.5 |
| | CML-HM[$\mathcal{S}, g_{\text{LGA}}$] | 32 | 39.6 | 50.6 | 23.9 | 51.5 | 22.1 | 23.0 | 23.7 | 15.6 | 0.5 | 0.5 | 27.0 | 28.7 | 26.8 | 9.2 | 47.3 |
| | CML-HM[$\mathcal{S}, g_{\text{LGA}}$]&ICS | 32 | 39.6 | 50.9 | 24.0 | 50.9 | 23.1 | 19.9 | 26.1 | 12.5 | 0.8 | 0.3 | 28.8 | 29.9 | 27.1 | 12.5 | 47.3 |
| CARS | N/A | N/A | 62.5 | 74.0 | 23.8 | 57.0 | 0.2 | 100.0 | 0.1 | 99.6 | 0.874 | 1.816 | 0.0 | 0.0 | 13.4 | 0.0 | 3.6 |
| | HM[$\mathcal{S}, g_{\text{LGA}}$]&ICS ($\epsilon = 0$) | 8 | 63.3 | 73.6 | 36.5 | 53.6 | 9.9 | 42.2 | 11.3 | 41.6 | 0.7 | 0.5 | 22.1 | 23.3 | 26.4 | 2.8 | 33.3 |
| | CML-EST[$\mathcal{R}$] | 8 | 52.0 | 64.1 | 26.6 | 47.1 | 12.2 | 51.8 | 13.6 | 40.8 | 0.6 | 0.8 | 17.9 | 14.9 | 19.5 | 0.8 | 30.8 |
| | CML-ACT[$\mathcal{R}$] | 8 | 51.5 | 62.7 | 27.4 | 49.1 | 23.7 | 23.4 | 24.0 | 19.5 | 0.4 | 0.7 | 22.8 | 22.1 | 25.6 | 2.6 | 45.5 |
| | CML-HM[$\mathcal{S}, g_{\text{LGA}}$] | 8 | 65.3 | 74.9 | 38.9 | 54.5 | 22.8 | 19.3 | 24.3 | 15.6 | 0.4 | 0.5 | 40.1 | 45.2 | 38.1 | 14.4 | 52.9 |
| | CML-HM[$\mathcal{S}, g_{\text{LGA}}$]&ICS | 8 | 64.5 | 75.1 | 38.7 | 53.9 | 23.5 | 16.3 | 26.6 | 12.1 | 0.7 | 0.3 | 45.8 | 48.6 | 41.1 | 18.8 | 54.0 |
| | HM[$\mathcal{S}, g_{\text{LGA}}$]&ICS ($\epsilon = 0$) | 32 | 62.0 | 72.7 | 34.7 | 51.5 | 11.3 | 38.1 | 12.4 | 36.7 | 0.7 | 0.5 | 25.1 | 27.9 | 28.4 | 4.7 | 36.3 |
| | CML-EST[$\mathcal{R}$] | 32 | 52.3 | 63.9 | 26.5 | 46.2 | 16.8 | 37.8 | 18.3 | 25.9 | 0.5 | 0.7 | 27.9 | 25.0 | 24.1 | 2.8 | 40.1 |
| | CML-ACT[$\mathcal{R}$] | 32 | 49.6 | 61.6 | 25.8 | 47.5 | 25.2 | 22.1 | 25.7 | 17.8 | 0.3 | 0.6 | 24.7 | 23.1 | 25.7 | 3.0 | 47.2 |
| | CML-HM[$\mathcal{S}, g_{\text{LGA}}$] | 32 | 63.8 | 74.4 | 38.0 | 54.0 | 24.4 | 17.3 | 25.4 | 13.4 | 0.4 | 0.4 | 43.4 | 48.1 | 40.1 | 17.8 | 55.5 |
| | CML-HM[$\mathcal{S}, g_{\text{LGA}}$]&ICS | 32 | 63.9 | 74.4 | 37.4 | 53.6 | 25.1 | 15.8 | 27.7 | 10.8 | 0.7 | 0.3 | 44.6 | 50.7 | 40.9 | 21.5 | 55.5 |
| SOP | N/A | N/A | 62.9 | 68.5 | 39.2 | 87.4 | 0.1 | 99.3 | 0.2 | 99.1 | 0.845 | 1.685 | 0.0 | 0.0 | 6.3 | 0.0 | 4.0 |
| | HM[$\mathcal{S}, g_{\text{LGA}}$] ($\epsilon = 0$) | N/A | 49.5 | 54.5 | 12.9 | 85.0 | 28.7 | 5.1 | 30.5 | 4.0 | 0.5 | 0.3 | 42.6 | 44.0 | 39.0 | 46.2 | 62.1 |
| | CML-EST[$\mathcal{R}$] | 8 | 54.4 | 60.1 | 15.4 | 85.8 | 14.6 | 25.9 | 14.8 | 21.8 | 0.4 | 0.7 | 22.9 | 21.2 | 26.4 | 12.6 | 42.4 |
| | CML-ACT[$\mathcal{R}$] | 8 | 50.2 | 55.6 | 13.6 | 85.3 | 26.7 | 9.2 | 27.7 | 7.0 | 0.3 | 0.4 | 29.2 | 31.4 | 30.1 | 23.4 | 55.8 |
| | CML-HM[$\mathcal{S}, g_{\text{LGA}}$] | 8 | 55.0 | 60.4 | 15.3 | 85.8 | 31.0 | 4.1 | 32.9 | 3.1 | 0.3 | 0.3 | 50.0 | 51.1 | 46.3 | 58.3 | 68.1 |
| | CML-HM[$\mathcal{S}, g_{\text{LGA}}$]&ICS | 8 | 49.3 | 54.6 | 12.9 | 85.1 | 35.6 | 3.0 | 38.3 | 2.0 | 0.8 | 0.1 | 46.2 | 47.7 | 43.0 | 63.0 | 65.3 |
| | HM[$\mathcal{S}, g_{\text{LGA}}$] ($\epsilon = 0$) | 32 | 47.2 | 52.0 | 12.0 | 84.7 | 29.6 | 4.5 | 31.8 | 3.6 | 0.5 | 0.3 | 41.4 | 42.9 | 37.9 | 46.9 | 62.2 |
| | CML-EST[$\mathcal{R}$] | 32 | 52.2 | 57.9 | 14.3 | 85.4 | 23.0 | 12.3 | 23.4 | 9.7 | 0.3 | 0.5 | 38.3 | 35.8 | 32.1 | 25.4 | 55.3 |
| | CML-ACT[$\mathcal{R}$] | 32 | 49.2 | 54.6 | 13.0 | 85.2 | 28.7 | 8.3 | 29.4 | 6.0 | 0.2 | 0.4 | 32.1 | 33.3 | 31.9 | 26.2 | 58.0 |
| | CML-HM[$\mathcal{S}, g_{\text{LGA}}$] | 32 | 54.3 | 59.7 | 15.0 | 85.7 | 31.3 | 3.9 | 33.1 | 3.0 | 0.3 | 0.3 | 51.1 | 51.7 | 46.6 | 59.9 | 68.7 |
| | CML-HM[$\mathcal{S}, g_{\text{LGA}}$]&ICS | 32 | 48.7 | 53.8 | 12.5 | 85.0 | 36.4 | 2.9 | 38.8 | 1.8 | 0.8 | 0.1 | 45.5 | 47.3 | 42.6 | 64.7 | 65.8 |

Table 2: Effect of robust initialization on state-of-the-art methods on three benchmark datasets: CUB, CARS, and SOP.

pre-training stage imbues the model with features robust agaisnt ranking attacks (ERS), while during metric learning stage, model significantly improves its recall ($\approx 10\%$ increase recall), with negligent loss in ranking robustness (ERS). Interestingly, even a small amount (small value of $\epsilon$) significantly improves the convergence of EST, making it much more competitive.

And while ACT and HM did not suffer catastrophic collapse like EST, the better shape of the metric space allowed for a significant improvement in learning, markedly improving recall and ranking robustness for all values of $\epsilon$ across all datasets.

### 4.6 OPTIMAL (INITIALIZATION) ROBUSTNESS LEVELS FOR METRIC LEARNING

From Figures 4 and 5, we can see every degree of initialization robustness ($\epsilon$) significantly improves ranking robustness (ERS) and recall. In general, we recommend $\epsilon = 4.0/255$ to be used for adversarial pre-training, it provides the best recall / robustness trade-off - robust pre-training with $\epsilon = 8.0/255$ achieves a small marginal improvement in ranking robustness (ERS) while suffering from large drop in performance on benign examples. For the best performance, depending on the application in mind one should run an ablation study with $\epsilon \in \{\frac{1}{255}, \frac{2}{255}, \frac{4}{255}\}$, and chose model which achieve ideal recall / robustness trade-off.

## 5 CONCLUSION

In our work, we take inspiration from curriculum learning and propose to learn features for robust retrieval by decomposing the learning process into two progressively harder sub-tasks: (1) learn robust features via adversarial-training followed by (2) robust metric learning to learn robust semantic features for accurate retrieval. We initially explore and quantify the adversarial ranking robustness of robust features. We show that robust models trained with adversarial training (against classification attacks) are already imbued with certain degree of robustness against adversarial ranking attacks (ARR), adding ARR as an additional benefit of robust representations. We show that initializing deep metric models with adversarialy robust models followed by adversarial ranking defenses significantly improves the model performance on retrieval of benign examples as well as their resistance to adversarial ranking attacks, achieving state of the art robust ranking performance across three benchmark datasets. We show that robust weight initialization improves model convergence for various sampling strategies, performance of various methods, and we show that these improvements are consistent across various benchmark datasets.

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

## A    APPENDIX

### A.1    ADDITIONAL ANALYSES

**Effect of triplet sampling strategies on ranking defenses**    Previous works (Roth et al., 2020; Liu et al., 2022) suggest that Soft-hard sampling ($\mathcal{S}$) (Roth et al., 2020) should achieve better Recall@1 as compared to random ($\mathcal{R}$) sampling. And although recall is very similar for EST ( for most $\epsilon$) and HM, we see a significant drop in recall for the Anti-Collapse Triplet (ACT) defense when the Soft-hard sampling is used to generate triplets. We hypothesize the Random sampling strategy outperforms the Soft-hard sampling strategy due to the conflict of how Soft-hard sampling generates triplets and the goal of the adversary in ACT. Soft-hard sampling generates triplets that "likely" have hard positives and hard negatives (see Supplement of Liu et al. (2022)), and as a result it is more likely that the positive and negative samples for a given anchor are further apart, making them harder to "collapse"[3].

### A.2    LIMITATIONS OF OUR WORK

Though our empirical results at the moment lack theoretical justification, we present strong empirical results which are consistent across multiple datasets and defenses and provide a evidence for benefit of initializing robust ranking defenses with robust features. Given the marked improvement of both retrieval performance and resistance to ranking attacks, we believe the empirical results are interesting in their own right and provide future avenues of research to the community.

### A.3    DISCUSSION OF SOCIETAL IMPACT

Though adversarial attacks pose profound risk for various scenarios, our work aims to alleviate such risk by identifying methods to improves adversarial robustness of real work systems. By addressing vulnerabilities in AI systems, this work contributes to the responsible and beneficial deployment of AI technologies. It helps create a more secure, reliable, and equitable environment for individuals, organizations, and society as a whole.

### A.4    USE OF EXISTING ASSETS

These experiments use publicly available code. All of the experiments are run with default parameters found in the corresponding repositories. The GitHub repositories and the licenses can be found at:

- The robust-models-transfer (**?**) repository is released under MIT license and contains the $l_\infty$ robust models used for robust weight initialization in our experiments. The repository can be found at
  `https://github.com/Microsoft/robust-models-transfer`
- RobRank (Zhou et al., 2021) repository us released under Apache (Version 2.0) license and can be found at
  `https://github.com/cdluminate/robrank`

All of the experiments are run on publicly available datasets

- The CUB-200-2011 (Welinder et al., 2010) dataset can be found at
  `http://www.vision.caltech.edu/datasets/cub_200_2011/`
- The CARS (Krause et al., 2013) dataset can be found at
  `http://ai.stanford.edu/~jkrause/cars/car_dataset.html`
- The Stanford Online Products (SOP) (Oh Song et al., 2016) dataset can be found at
  `https://cvgl.stanford.edu/projects/lifted_struct/`

---

[3]The goal of ACT defense is to separate "collapsed" embeddings of the positive and negative samples Zhou & Patel (2022).

## A.5 Hardware Configuration

Our code is developed on an internal cluster, where each server node is equipped with 4 NVIDIA Tesla A100 cards (each with 40 GB of VRAM), paired with a 64-core AMD EPYC cpu and 256GB of memory. All of our experiments utilize the ResNet-18 architecture.

## A.6 Adversarial ranking attack: Candidate attack (CA)

Consider the case of a Candidate attack (CA) Zhou et al. (2020a), where you aim to raise or lower the rank of single candidate, $c$, with respect to a set of Queries, $Q = \{q_1, \ldots, q_w\}$. Note that locally in DML, a candidate, $c_p$, is ranked ahead of $c_n$ if it is closer to the query image, i.e. $d(q, c_p) < d(q, c_n)$. Such rank ordering can also be formulated in the form of a triplet loss

$$L_{trip}(q, c_p, c_n) = \max(0, \beta + d(q, c_p) - d(q, c_n)). \tag{3}$$

Note that a CA that raises the rank of c with respect to every query $q \in Q$ ahead of all the candidates (C) can be formulated as a series of inequalities and subsequently a sum of triplet losses,

$$L_{CA+}(c, Q; X) = \sum_{q \in Q} \sum_{x \in X} L_{trip}(q, c, x). \tag{4}$$

Zhou et al. (2020a) point out that the attack / perturbed image is a solution to the following constrained optimization problem:

$$r^* = \arg\min_{r \in \Gamma} L_{CA+}(c + r, Q; X). \tag{5}$$

For a more detailed description of adversarial ranking attacks, see Zhou et al. (2020a).

## A.7 Generating adversarial examples for the EST defense.

This is done by running a $k$-step PGD attacks against a metric loss (e.g., cosine similarity or euclidean distance). Let $x_{orig}$ be a copy of the original image. To find the shifted adversarial examples, the image $x$ is first perturbed in its $\epsilon$-neighborhood, i.e.

$$x_0 \coloneqq x + U(-\epsilon, \epsilon), \tag{6}$$

where $U$ is a uniform distribution. It then follows the generation process of the Basic Iterative Method Kurakin et al. (2016),

$$x^{k+1} \coloneqq \Pi_S \left( x_k + \alpha \cdot \text{sign} \left( \nabla_x \mathcal{L}_{\text{metric}}(x_k, x_{orig}; F_\theta) \right) \right) \tag{7}$$

where $K$ is the number of "attacks steps", or iterations of projected gradient descent performed to find the worst case $l_p$ bounded adversarial example $x^{\text{adv}}$. $\Pi_p$ is an operator that will project the iterates onto an $l_p$ ball, where in our case $p = 2$ or $\infty$.

## A.8 Further experimental results

Tables A5 and A1 are summarized in Figure 4. Table A3 shows the detailed model performance results for Table A1.

## A.9 Empirical robustness score - ERS

The ensemble of adversarial rankings attacks to quantify ranking robustness include the following attacks:

(a) Candidate attacks ($CA+$, $CA-$) (Zhou et al., 2020a) aim is to increase or lower ranks of chosen candidate $c$ with respect to a query set $Q$

(b) Query attacks ($QA+$, $QA-$) (Zhou et al., 2020a) aim to raise or lower the rank of a set of candidates $C$ by perturbing a single query $q$

(c) Targeted Mismatch Attack (TMA)(Tolias et al., 2019) increases the similarity between two arbitrary samples

| $H_S$ / $H_D$ | Random [$\mathcal{R}$] | | Semihard [$\mathcal{M}$] | | Softhard [$\mathcal{S}$] | | Distance [$\mathcal{D}$] | | Hardest [$\mathcal{H}$] | | $g_{LGA}$ | |
|---|---|---|---|---|---|---|---|---|---|---|---|---|
| | R@1 | ERS | R@1 | ERS | R@1 | ERS | R@1 | ERS | R@1 | ERS | R@1 | ERS |
| Random [$\mathcal{R}$] | 42.5 | 15.1 | 31.1 | 45.5 | 30.6 | 49.4 | 28.9 | 50.3 | 29.6 | 50.6 | 31.0 | 47.7 |
| Semihard [$\mathcal{M}$] | 30.0 | 12.5 | 28.6 | 9.4 | 39.0 | 42.4 | 32.6 | 49.9 | 32.9 | 50.0 | 34.6 | 40.7 |
| Softhard [$\mathcal{S}$] | 45.7 | 36.9 | 41.7 | 44.6 | 48.3 | 13.0 | 20.8 | 41.1 | 20.1 | 41.8 | 40.4 | 47.2 |
| Distance [$\mathcal{D}$] | 47.5 | 10.0 | 47.8 | 14.8 | 23.0 | 26.1 | 47.9 | 9.9 | 2.3 | 27.9 | 42.5 | 34.4 |
| Hardest [$\mathcal{H}$] | 47.4 | 10.0 | 48.3 | 15.0 | 22.9 | 26.4 | 2.3 | 27.7 | 47.7 | 9.5 | 42.5 | 34.4 |

Table A1: Clean (R@1) and robust (ERS) performance (for the CUB Dataset) of ResNet-18 models initialized with a $l_\infty$ robust ImageNet checkpoint ($\epsilon = 4.0/255$) for different combinations of source & destination hardness sampling strategies. Models on the diagonal are regularly (instead of adversarially) trained, and the last-epoch performance is reported to stay consistent with Zhou & Patel (2022). Last columns shows the effect of robust initialization on gradual adversary as $H_D$ in hardness manipulation.

| Dataset | Defense | $\epsilon$ | Benign Example | | | | White-Box Attacks for Robustness Evaluation | | | | | | | | | | ERS↑ |
|---|---|---|---|---|---|---|---|---|---|---|---|---|---|---|---|---|---|
| | | | R@1↑ | R@2↑ | mAP↑ | NMI↑ | CA+↑ | CA-↓ | QA+↑ | QA-↓ | TMA↓ | ES:D↓ | ES:R↑ | LTM↑ | GTM↑ | GTT↑ | |
| CUB | HM[$\mathcal{R}, g_{LGA}$] | 4.0 | 31.0 | 40.5 | 13.4 | 44.2 | 27.7 | 25.0 | 31.6 | 15.8 | 0.4 | 0.6 | 19.9 | 19.3 | 20.3 | 8.0 | 47.7 |
| | HM[$\mathcal{M}, g_{LGA}$] | 4.0 | 34.6 | 44.9 | 16.7 | 48.7 | 20.3 | 31.8 | 21.5 | 25.4 | 0.4 | 0.8 | 18.5 | 16.8 | 20.9 | 3.3 | 40.7 |
| | HM[$\mathcal{S}, g_{LGA}$] | 4.0 | 40.4 | 51.3 | 18.6 | 51.3 | 21.9 | 23.3 | 23.9 | 15.4 | 0.5 | 0.5 | 28.1 | 28.8 | 26.4 | 9.5 | 47.2 |
| | HM[$\mathcal{D}, g_{LGA}$] | 4.0 | 42.5 | 52.8 | 19.9 | 51.3 | 11.8 | 44.3 | 10.9 | 41.0 | 0.4 | 0.9 | 21.1 | 18.0 | 23.4 | 2.8 | 34.4 |
| | HM[$\mathcal{H}, g_{LGA}$] | 4.0 | 42.5 | 52.5 | 19.3 | 50.9 | 12.0 | 43.9 | 11.3 | 41.5 | 0.4 | 0.9 | 21.0 | 18.0 | 23.5 | 2.6 | 34.5 |

Table A2: Effect of robust initialization on gradual adversary as $H_D$ in hardness manipulation.. The "↑" mark means "the higher the better", while "↓" means the opposite.

(d) Embedding Shift attack (ES)(Feng et al., 2020) moves the embedding of a query as far as possible from its original position

(e) Learning-To-Misrank attack (LTM) (Wang et al., 2020) aims to perturb rank ordering by minimizing / maximizing distance of matched / unmatched pairs

(f) Greedy Top-1 Misranking attack (GTM) (Zhou et al., 2021) reduces the distance between adversarial query and closest non-matching candidate, and

(g) Greedy Top-1 Translocation attack (GTT) (Zhou et al., 2021) moves the top retrieval result of out top-$k$ ranked items.

| Dataset | Defense | $\eta$ | Benign Example | | | | White-Box Attacks for Robustness Evaluation | | | | | | | | | | ERS |
|---|---|---|---|---|---|---|---|---|---|---|---|---|---|---|---|---|---|
| | | | R@1↑ | R@2↑ | mAP↑ | NMI↑ | CA+↑ | CA-↓ | QA+↑ | QA-↓ | TMA↓ | ES:D↓ | ES:R↑ | LTM↑ | GTM↑ | GTT↑ | |
| CUB | HM[$\mathcal{R}$,$\mathcal{R}$] | N/A | 42.5 | 53.8 | 24.6 | 52.2 | 4.4 | 76.7 | 3.3 | 76.6 | 0.7 | 1.2 | 4.5 | 2.9 | 13.9 | 0.2 | 15.1 |
| | HM[$\mathcal{R}$,$\mathcal{M}$] | 8 | 31.1 | 40.2 | 17.1 | 45.3 | 26.7 | 29.5 | 29.4 | 19.0 | 0.4 | 0.6 | 19.6 | 19.0 | 19.9 | 6.2 | 45.5 |
| | HM[$\mathcal{R}$,$\mathcal{S}$] | 8 | 30.6 | 40.1 | 16.5 | 44.8 | 29.3 | 20.9 | 32.7 | 11.5 | 0.6 | 0.4 | 22.0 | 22.4 | 21.6 | 12.6 | 49.4 |
| | HM[$\mathcal{R}$,$\mathcal{D}$] | 8 | 28.9 | 38.9 | 15.2 | 43.1 | 30.9 | 17.5 | 35.0 | 9.5 | 0.7 | 0.3 | 21.5 | 21.9 | 21.4 | 16.2 | 50.3 |
| | HM[$\mathcal{R}$,$\mathcal{H}$] | 8 | 29.6 | 39.8 | 15.7 | 43.4 | 31.2 | 17.1 | 35.1 | 9.2 | 0.6 | 0.3 | 21.6 | 22.1 | 21.5 | 16.5 | 50.6 |
| CUB | HM[$\mathcal{M}$,$\mathcal{R}$] | 8 | 30.0 | 41.1 | 16.8 | 46.1 | 3.9 | 88.1 | 5.0 | 85.1 | 0.7 | 1.4 | 3.3 | 0.8 | 11.5 | 0.0 | 12.5 |
| | HM[$\mathcal{M}$,$\mathcal{M}$] | N/A | 28.6 | 39.5 | 15.8 | 45.5 | 2.9 | 95.2 | 3.4 | 92.2 | 0.7 | 1.5 | 1.2 | 0.3 | 10.8 | 0.0 | 9.4 |
| | HM[$\mathcal{M}$,$\mathcal{S}$] | 8 | 39.0 | 49.5 | 22.6 | 49.5 | 19.0 | 28.1 | 20.8 | 22.4 | 0.5 | 0.6 | 25.2 | 21.7 | 24.2 | 6.2 | 42.4 |
| | HM[$\mathcal{M}$,$\mathcal{D}$] | 8 | 32.6 | 42.6 | 18.2 | 44.9 | 28.2 | 13.9 | 33.2 | 7.8 | 0.8 | 0.3 | 25.8 | 25.3 | 23.2 | 16.6 | 49.9 |
| | HM[$\mathcal{M}$,$\mathcal{H}$] | 8 | 32.9 | 42.7 | 18.5 | 44.8 | 28.6 | 14.1 | 33.0 | 7.9 | 0.8 | 0.3 | 26.4 | 25.0 | 22.9 | 16.9 | 50.0 |
| CUB | HM[$\mathcal{S}$,$\mathcal{R}$] | 8 | 45.7 | 57.4 | 28.6 | 54.9 | 12.8 | 37.2 | 13.7 | 31.1 | 0.7 | 0.5 | 26.1 | 26.1 | 25.4 | 4.1 | 36.9 |
| | HM[$\mathcal{S}$,$\mathcal{M}$] | 8 | 41.7 | 53.6 | 25.7 | 52.5 | 20.0 | 26.6 | 22.2 | 19.2 | 0.6 | 0.5 | 27.9 | 28.1 | 25.7 | 7.1 | 44.6 |
| | HM[$\mathcal{S}$,$\mathcal{S}$] | N/A | 48.3 | 60.5 | 29.6 | 55.9 | 1.2 | 86.3 | 1.2 | 84.5 | 0.9 | 0.7 | 5.3 | 3.4 | 15.1 | 0.0 | 13.0 |
| | HM[$\mathcal{S}$,$\mathcal{D}$] | 8 | 20.8 | 30.2 | 6.4 | 33.2 | 23.1 | 36.2 | 32.6 | 21.0 | 1.0 | 0.0 | 12.0 | 19.0 | 18.3 | 7.5 | 41.1 |
| | HM[$\mathcal{S}$,$\mathcal{H}$] | 8 | 20.1 | 29.5 | 6.5 | 34.4 | 24.0 | 32.7 | 32.6 | 18.3 | 1.0 | 0.0 | 13.7 | 17.4 | 17.7 | 7.6 | 41.8 |
| CUB | HM[$\mathcal{D}$,$\mathcal{R}$] | 8 | 47.5 | 59.1 | 28.8 | 55.2 | 1.1 | 91.8 | 0.8 | 92.7 | 0.7 | 1.4 | 3.1 | 1.9 | 14.5 | 0.0 | 10.0 |
| | HM[$\mathcal{D}$,$\mathcal{M}$] | 8 | 547.8 | 59.7 | 30.1 | 55.8 | 2.4 | 80.8 | 1.6 | 83.5 | 0.7 | 1.1 | 7.0 | 4.6 | 17.0 | 0.0 | 14.8 |
| | HM[$\mathcal{D}$,$\mathcal{S}$] | 8 | 23.0 | 32.9 | 8.6 | 35.7 | 6.5 | 75.6 | 16.3 | 46.3 | 1.0 | 0.0 | 10.7 | 9.4 | 14.3 | 2.5 | 26.1 |
| | HM[$\mathcal{D}$,$\mathcal{D}$] | N/A | 47.9 | 58.8 | 29.3 | 54.4 | 1.1 | 91.7 | 0.6 | 93.2 | 0.7 | 1.4 | 3.0 | 1.7 | 14.5 | 0.0 | 9.9 |
| | HM[$\mathcal{D}$,$\mathcal{H}$] | 8 | 2.3 | 3.1 | 1.2 | 7.7 | 14.8 | 86.8 | 39.9 | 52.2 | 1.0 | 0.0 | 1.5 | 1.6 | 1.6 | 3.8 | 27.9 |
| CUB | HM[$\mathcal{H}$,$\mathcal{R}$] | 8 | 47.4 | 58.9 | 29.1 | 54.4 | 1.1 | 91.8 | 0.8 | 92.9 | 0.7 | 1.4 | 3.0 | 2.0 | 14.6 | 0.1 | 10.0 |
| | HM[$\mathcal{H}$,$\mathcal{M}$] | 8 | 48.3 | 59.7 | 30.3 | 55.8 | 2.6 | 80.2 | 1.9 | 82.2 | 0.7 | 1.1 | 7.0 | 4.7 | 17.0 | 0.1 | 15.0 |
| | HM[$\mathcal{H}$,$\mathcal{S}$] | 8 | 22.9 | 33.0 | 8.2 | 33.9 | 6.5 | 76.2 | 17.2 | 46.3 | 1.0 | 0.0 | 12.9 | 10.1 | 13.7 | 2.3 | 26.4 |
| | HM[$\mathcal{H}$,$\mathcal{D}$] | 8 | 2.3 | 3.0 | 1.2 | 9.4 | 14.6 | 85.7 | 38.9 | 52.8 | 1.0 | 0.0 | 1.9 | 1.2 | 1.2 | 4.0 | 27.7 |
| | HM[$\mathcal{H}$,$\mathcal{H}$] | N/A | 47.7 | 60.0 | 29.4 | 55.3 | 0.9 | 93.0 | 0.6 | 94.2 | 0.7 | 1.4 | 2.9 | 1.4 | 14.5 | 0.0 | 9.5 |

Table A3: Detailed model performance for all combinations of source and destination sampling strategies in Table 2. in the Paper. The symbols $\mathcal{R}, \mathcal{M}, \mathcal{S}, \mathcal{D}, \mathcal{H}$ denote Random, Semihard, Softhard, Distance and Hardest triplet sampling strategies, respectively.

| Defense | $\eta$ | $\epsilon$ | Benign Example | | | | White-Box Attacks for Robustness Evaluation | | | | | | | | | | ERS↑ |
|---|---|---|---|---|---|---|---|---|---|---|---|---|---|---|---|---|---|
| | | | R@1↑ | R@2↑ | mAP↑ | NMI↑ | CA+↑ | CA-↓ | QA+↑ | QA-↓ | TMA↓ | ES:D↓ | ES:R↑ | LTM↑ | GTM↑ | GTT↑ | |
| N/A[$\mathcal{R}$] | N/A | N/A | 53.9 | 66.4 | 26.1 | 59.5 | 0.0 | 100.0 | 0.0 | 99.9 | 0.883 | 1.762 | 0.0 | 0.0 | 14.1 | 0.0 | 3.8 |
| EST[$\mathcal{R}$] | 8 | 0.0 | 35.0 | 46.1 | 17.6 | 45.0 | 0.4 | 97.8 | 0.4 | 93.3 | 0.9 | 1.2 | 5.1 | 0.3 | 10.5 | 0.0 | 7.3 |
| | 8 | 1.0 | 44.3 | 56.0 | 26.0 | 53.2 | 8.2 | 58.6 | 7.1 | 51.1 | 0.6 | 0.9 | 18.8 | 11.7 | 16.9 | 0.4 | 26.1 |
| | 8 | 4.0 | 37.9 | 49.6 | 22.5 | 50.4 | 16.1 | 36.7 | 17.0 | 25.8 | 0.5 | 0.6 | 25.2 | 22.3 | 20.9 | 3.6 | 39.3 |
| | 32 | 0.0 | 7.0 | 11.2 | 2.0 | 25.5 | 1.5 | 98.7 | 0.8 | 98.0 | 0.8 | 1.6 | 1.0 | 0.0 | 3.9 | 0.0 | 4.9 |
| | 32 | 1.0 | 42.2 | 52.7 | 24.6 | 50.7 | 15.5 | 38.9 | 15.3 | 28.7 | 0.5 | 0.7 | 26.0 | 20.5 | 19.9 | 2.1 | 37.5 |
| | 32 | 4.0 | 37.0 | 49.0 | 21.0 | 49.6 | 19.8 | 36.6 | 22.3 | 18.2 | 0.5 | 0.6 | 26.9 | 25.3 | 21.6 | 4.4 | 42.7 |
| ACT[$\mathcal{R}$] | 8 | 0.0 | 31.0 | 41.4 | 16.8 | 45.4 | 13.3 | 47.3 | 12.3 | 41.8 | 0.6 | 0.9 | 11.3 | 7.6 | 14.3 | 0.4 | 29.4 |
| | 8 | 1.0 | 41.9 | 53.3 | 25.3 | 51.1 | 17.7 | 36.6 | 16.4 | 29.3 | 0.4 | 0.8 | 21.4 | 18.6 | 22.0 | 2.2 | 38.2 |
| | 8 | 4.0 | 36.2 | 47.6 | 22.6 | 49.7 | 20.6 | 30.0 | 21.3 | 22.1 | 0.4 | 0.7 | 19.4 | 20.6 | 21.5 | 4.2 | 42.1 |
| | 32 | 0.0 | 26.4 | 36.1 | 13.6 | 42.6 | 17.0 | 38.2 | 17.2 | 29.7 | 0.5 | 0.8 | 13.4 | 8.3 | 14.0 | 0.8 | 34.5 |
| | 32 | 1.0 | 40.7 | 50.7 | 24.2 | 50.7 | 19.1 | 33.0 | 19.0 | 25.6 | 0.4 | 0.7 | 23.1 | 19.7 | 21.5 | 2.7 | 40.5 |
| | 32 | 4.0 | 36.6 | 46.5 | 21.6 | 48.9 | 21.1 | 28.4 | 21.6 | 21.3 | 0.4 | 0.7 | 19.8 | 19.3 | 21.3 | 3.9 | 42.5 |
| HM[$\mathcal{S}, g_{\text{LGA}}$] | 8 | 0.0 | 39.5 | 49.8 | 23.8 | 50.2 | 10.5 | 51.0 | 10.5 | 48.7 | 0.6 | 0.8 | 12.6 | 11.0 | 17.3 | 0.9 | 28.1 |
| | 8 | 1.0 | 43.8 | 54.8 | 26.6 | 53.9 | 18.4 | 31.0 | 19.2 | 23.6 | 0.5 | 0.6 | 26.6 | 26.0 | 25.9 | 4.1 | 42.0 |
| | 8 | 4.0 | 42.2 | 53.1 | 25.1 | 51.6 | 20.8 | 25.9 | 21.8 | 18.3 | 0.5 | 0.5 | 27.6 | 28.2 | 25.5 | 8.0 | 45.2 |
| | 32 | 0.0 | 37.6 | 48.5 | 22.2 | 48.5 | 12.4 | 47.1 | 12.3 | 43.1 | 0.6 | 0.8 | 14.7 | 11.8 | 18.3 | 1.0 | 30.7 |
| | 32 | 1.0 | 43.3 | 53.5 | 25.9 | 52.4 | 19.7 | 27.7 | 20.5 | 20.5 | 0.5 | 0.6 | 25.8 | 26.9 | 26.0 | 5.5 | 43.9 |
| | 32 | 4.0 | 39.6 | 50.6 | 23.9 | 51.5 | 22.1 | 23.0 | 23.7 | 15.6 | 0.5 | 0.5 | 27.0 | 28.7 | 26.8 | 9.2 | 47.3 |
| HM[$\mathcal{S}, g_{\text{LGA}}$]&ICS | 8 | 0.0 | 38.1 | 48.9 | 22.7 | 49.8 | 11.3 | 47.8 | 11.8 | 43.2 | 0.8 | 0.5 | 14.7 | 12.3 | 18.2 | 0.9 | 29.1 |
| | 8 | 1.0 | 44.1 | 55.8 | 26.3 | 51.9 | 19.3 | 27.1 | 22.2 | 19.5 | 0.8 | 0.4 | 28.0 | 27.6 | 25.5 | 5.9 | 42.6 |
| | 8 | 4.0 | 41.1 | 51.9 | 25.1 | 51.8 | 21.8 | 21.3 | 25.5 | 14.9 | 0.8 | 0.3 | 29.0 | 30.0 | 26.1 | 10.4 | 46.0 |
| | 32 | 0.0 | 36.8 | 47.8 | 21.3 | 48.7 | 13.4 | 44.9 | 13.9 | 38.8 | 0.8 | 0.6 | 15.1 | 13.0 | 18.7 | 1.3 | 31.3 |
| | 32 | 1.0 | 42.7 | 54.2 | 25.7 | 51.7 | 20.2 | 25.0 | 21.9 | 17.3 | 0.8 | 0.4 | 27.1 | 29.1 | 27.1 | 6.8 | 43.8 |
| | 32 | 4.0 | 39.6 | 50.9 | 24.0 | 50.9 | 23.1 | 19.9 | 26.1 | 12.5 | 0.8 | 0.3 | 28.8 | 29.9 | 27.1 | 12.5 | 47.3 |

Table A4: Effect of $\eta$ and $\epsilon$ on state-of-the-art defenses: a) Embedding-Shifted Triplet (EST) (Zhou et al., 2020a), (b) Anti-Collapse Triplet (ACT) (Zhou et al., 2021), and (c) Hardness Manipulations (HM) (Zhou & Patel, 2022).

| Dataset | Defense | $\epsilon$ | Benign Example | | | | White-Box Attacks for Robustness Evaluation | | | | | | | | | | ERS↑ |
|---|---|---|---|---|---|---|---|---|---|---|---|---|---|---|---|---|---|
| | | | R@1↑ | R@2↑ | mAP↑ | NMI↑ | CA+↑ | CA-↓ | QA+↑ | QA-↓ | TMA↓ | ES:D↓ | ES:R↑ | LTM↑ | GTM↑ | GTT↑ | |
| CUB | N/A[$\mathcal{R}$] | N/A | 53.9 | 66.4 | 26.1 | 59.5 | 0.0 | 100.0 | 0.0 | 99.9 | 0.883 | 1.762 | 0.0 | 0.0 | 14.1 | 0.0 | 3.8 |
| CUB | EST[$\mathcal{R}$] | 0.0 | 35.0 | 46.1 | 17.6 | 45.0 | 0.4 | 97.8 | 0.4 | 93.3 | 0.9 | 1.2 | 5.1 | 0.3 | 10.5 | 0.0 | 7.3 |
| | EST[$\mathcal{R}$] | 0.5 | 45.4 | 57.6 | 27.6 | 53.8 | 7.3 | 60.7 | 5.7 | 59.9 | 0.6 | 1.0 | 15.1 | 9.3 | 16.5 | 0.3 | 23.5 |
| | EST[$\mathcal{R}$] | 1.0 | 44.3 | 56.0 | 26.0 | 53.2 | 8.2 | 58.6 | 7.1 | 51.1 | 0.6 | 0.9 | 18.8 | 11.7 | 16.9 | 0.4 | 26.1 |
| | EST[$\mathcal{R}$] | 2.0 | 40.9 | 53.1 | 23.9 | 51.2 | 12.0 | 46.4 | 12.2 | 35.2 | 0.6 | 0.7 | 24.2 | 18.2 | 19.6 | 1.3 | 33.6 |
| | EST[$\mathcal{R}$] | 4.0 | 37.9 | 49.6 | 22.5 | 50.4 | 16.1 | 36.7 | 17.0 | 25.8 | 0.5 | 0.6 | 25.2 | 22.3 | 20.9 | 3.6 | 39.3 |
| | EST[$\mathcal{R}$] | 8.0 | 33.7 | 44.1 | 18.4 | 46.9 | 16.6 | 38.0 | 17.4 | 27.0 | 0.5 | 0.6 | 22.7 | 19.9 | 19.4 | 3.6 | 38.5 |
| CUB | EST[$\mathcal{S}$] | 0.0 | 43.0 | 54.7 | 24.6 | 50.7 | 0.0 | 99.8 | 0.1 | 97.9 | 1.0 | 0.5 | 7.8 | 0.2 | 11.0 | 0.0 | 9.8 |
| | EST[$\mathcal{S}$] | 0.5 | 49.5 | 61.0 | 30.8 | 55.5 | 3.0 | 70.6 | 3.7 | 66.0 | 0.9 | 0.5 | 22.6 | 9.5 | 18.6 | 0.2 | 21.3 |
| | EST[$\mathcal{S}$] | 1.0 | 48.7 | 60.3 | 29.8 | 56.4 | 3.7 | 66.6 | 4.5 | 59.5 | 0.9 | 0.4 | 22.6 | 15.8 | 20.4 | 0.4 | 23.7 |
| | EST[$\mathcal{S}$] | 2.0 | 45.6 | 57.8 | 27.9 | 54.6 | 7.6 | 55.6 | 9.0 | 43.1 | 0.9 | 0.4 | 27.4 | 24.5 | 21.9 | 1.1 | 30.0 |
| | EST[$\mathcal{S}$] | 4.0 | 41.3 | 53.1 | 24.6 | 52.5 | 9.0 | 46.9 | 11.1 | 35.2 | 0.9 | 0.4 | 27.9 | 21.7 | 22.1 | 2.3 | 32.7 |
| | EST[$\mathcal{S}$] | 8.0 | 39.3 | 50.3 | 22.1 | 50.2 | 12.8 | 39.0 | 15.9 | 27.3 | 0.9 | 0.3 | 27.3 | 25.1 | 21.9 | 3.6 | 36.6 |
| CUB | ACT[$\mathcal{R}$] | 0.0 | 31.0 | 41.4 | 16.8 | 45.4 | 13.3 | 47.3 | 12.3 | 41.8 | 0.6 | 0.9 | 11.3 | 7.6 | 14.3 | 0.4 | 29.4 |
| | ACT[$\mathcal{R}$] | 0.5 | 42.0 | 52.7 | 25.7 | 51.7 | 16.4 | 40.0 | 15.2 | 32.7 | 0.5 | 0.8 | 17.6 | 16.1 | 20.5 | 1.7 | 35.6 |
| | ACT[$\mathcal{R}$] | 1.0 | 41.9 | 53.3 | 25.3 | 51.1 | 17.7 | 36.6 | 16.4 | 29.3 | 0.4 | 0.8 | 21.4 | 18.6 | 22.0 | 2.2 | 38.2 |
| | ACT[$\mathcal{R}$] | 2.0 | 40.1 | 50.5 | 24.1 | 51.2 | 18.4 | 34.0 | 17.9 | 26.0 | 0.4 | 0.8 | 20.1 | 19.6 | 21.0 | 3.1 | 39.4 |
| | ACT[$\mathcal{R}$] | 4.0 | 36.2 | 47.6 | 22.6 | 49.7 | 20.6 | 30.0 | 21.3 | 22.1 | 0.4 | 0.7 | 19.4 | 20.6 | 21.5 | 4.2 | 42.1 |
| | ACT[$\mathcal{R}$] | 8.0 | 32.8 | 43.1 | 18.5 | 46.8 | 20.6 | 30.1 | 21.2 | 22.2 | 0.4 | 0.7 | 18.1 | 17.8 | 20.6 | 3.9 | 41.5 |
| CUB | ACT[$\mathcal{S}$] | 0.0 | 45.3 | 56.6 | 28.6 | 54.6 | 2.7 | 80.8 | 1.8 | 83.2 | 0.7 | 1.3 | 4.8 | 3.6 | 14.4 | 0.0 | 13.7 |
| | ACT[$\mathcal{S}$] | 0.5 | 52.8 | 63.6 | 34.2 | 59.2 | 6.7 | 59.9 | 5.3 | 59.9 | 0.6 | 1.0 | 12.5 | 12.1 | 19.5 | 0.3 | 23.7 |
| | ACT[$\mathcal{S}$] | 1.0 | 50.8 | 62.1 | 32.9 | 59.2 | 8.3 | 56.8 | 6.3 | 54.4 | 0.6 | 0.9 | 17.3 | 15.7 | 20.8 | 0.5 | 26.7 |
| | ACT[$\mathcal{S}$] | 2.0 | 48.5 | 60.3 | 31.1 | 57.7 | 9.5 | 49.6 | 7.9 | 47.9 | 0.6 | 0.9 | 19.2 | 17.3 | 22.1 | 0.9 | 29.6 |
| | ACT[$\mathcal{S}$] | 4.0 | 45.0 | 57.1 | 28.7 | 55.3 | 10.4 | 46.4 | 9.4 | 43.8 | 0.6 | 0.8 | 19.0 | 18.2 | 22.1 | 1.3 | 31.2 |
| | ACT[$\mathcal{S}$] | 8.0 | 41.6 | 53.8 | 25.4 | 52.3 | 11.5 | 43.9 | 10.9 | 39.7 | 0.6 | 0.7 | 19.0 | 17.4 | 21.2 | 1.9 | 32.4 |
| CUB | HM[$\mathcal{R}, g_{\text{LGA}}$] | 0.0 | 27.3 | 36.8 | 13.9 | 43.2 | 19.2 | 47.3 | 21.6 | 34.4 | 0.5 | 0.9 | 10.3 | 8.2 | 14.5 | 1.2 | 33.7 |
| | HM[$\mathcal{R}, g_{\text{LGA}}$] | 0.5 | 33.9 | 43.7 | 18.9 | 46.1 | 22.9 | 36.9 | 25.3 | 25.6 | 0.5 | 0.8 | 16.5 | 14.6 | 20.1 | 3.4 | 40.4 |
| | HM[$\mathcal{R}, g_{\text{LGA}}$] | 1.0 | 33.9 | 43.6 | 18.6 | 46.2 | 24.3 | 33.7 | 26.9 | 22.6 | 0.4 | 0.7 | 19.8 | 17.5 | 20.2 | 4.6 | 43.0 |
| | HM[$\mathcal{R}, g_{\text{LGA}}$] | 2.0 | 31.6 | 41.8 | 17.5 | 45.3 | 25.5 | 32.1 | 27.7 | 20.6 | 0.4 | 0.6 | 20.3 | 18.1 | 20.2 | 5.6 | 44.2 |
| | HM[$\mathcal{R}, g_{\text{LGA}}$] | 4.0 | 31.0 | 40.7 | 16.7 | 45.6 | 27.7 | 26.6 | 30.1 | 16.6 | 0.4 | 0.6 | 19.6 | 18.7 | 20.0 | 6.9 | 46.7 |
| | HM[$\mathcal{R}, g_{\text{LGA}}$] | 8.0 | 26.9 | 35.6 | 13.7 | 42.9 | 28.4 | 23.9 | 32.4 | 15.3 | 0.4 | 0.6 | 18.2 | 17.3 | 19.0 | 9.7 | 47.8 |
| CUB | HM[$\mathcal{S}, g_{\text{LGA}}$] | 0.0 | 39.5 | 49.8 | 23.8 | 50.2 | 10.5 | 51.0 | 10.5 | 48.7 | 0.6 | 0.8 | 12.6 | 11.0 | 17.3 | 0.9 | 28.1 |
| | HM[$\mathcal{S}, g_{\text{LGA}}$] | 0.5 | 44.6 | 55.5 | 27.7 | 53.2 | 17.2 | 33.4 | 17.8 | 26.4 | 0.5 | 0.6 | 25.0 | 23.6 | 24.3 | 2.9 | 40.0 |
| | HM[$\mathcal{S}, g_{\text{LGA}}$] | 1.0 | 43.8 | 54.8 | 26.6 | 53.9 | 18.4 | 31.0 | 19.2 | 23.6 | 0.5 | 0.6 | 26.6 | 26.0 | 25.9 | 4.1 | 42.0 |
| | HM[$\mathcal{S}, g_{\text{LGA}}$] | 2.0 | 43.9 | 54.9 | 26.5 | 52.9 | 19.3 | 28.6 | 20.7 | 20.3 | 0.5 | 0.6 | 26.5 | 27.0 | 26.2 | 5.3 | 43.5 |
| | HM[$\mathcal{S}, g_{\text{LGA}}$] | 4.0 | 42.2 | 53.1 | 25.1 | 51.6 | 20.8 | 25.9 | 21.8 | 18.3 | 0.5 | 0.5 | 27.6 | 28.2 | 25.5 | 8.0 | 45.2 |
| | HM[$\mathcal{S}, g_{\text{LGA}}$] | 8.0 | 37.6 | 48.9 | 22.3 | 49.8 | 21.3 | 23.9 | 22.5 | 17.0 | 0.5 | 0.5 | 25.0 | 26.8 | 24.6 | 8.7 | 45.4 |

Table A5: Effect of robust initialization on ranking defenses for deep representations learning: Embedding-Shifted Triplet (EST), Anti-collapse triplet (ACT) and Hardness Manipulations (HM). The "↑" mark means "the higher the better", while "↓" means the opposite.

