# OpenReview forum: "Curriculum metric learning for robust image retrieval"
_ICLR.cc/2024/Conference — Submitted to ICLR 2024_

### Official Review · Reviewer_QZHk · 2023-10-30

**Soundness:** 3 good
**Presentation:** 3 good
**Contribution:** 2 fair
**Rating:** 3
**Confidence:** 3

**Summary:**

This paper investigates the robustness of deep metric learning models to adversarial ranking attacks. The authors first show that even simply initializing a retrieval model with an adversarially trained model on a classification task can greatly improve its robustness to adversarial ranking attacks. Based on this observation, the authors further argue that initializing deep metric models with adversarially trained ones and then performing adversarial ranking defense can further improve their resistance to adversarial ranking attacks. Experimental results on multiple datasets support the authors' findings.

**Strengths:**

Strengths:

The paper has a simple and straightforward motivation, with a detailed description of the approach that is easy to replicate, which makes it accessible to researchers and practitioners alike.

The effectiveness of the proposed method is demonstrated through experimental results, showing significant improvements over baseline methods with simple modifications. Besides, the authors provide experimental results on multiple datasets, which increases the robustness of their findings and reinforces the significance of their approach. This adds credibility to the research and strengthens the impact of the results.

This paper addresses an important problem in deep metric learning and explores the robustness of retrieval models against adversarial ranking attacks.

**Weaknesses:**

Weaknesses:

Lack of novelty: The paper's main finding, that using adversarial trained models to initialize retrieval models improves their robustness against adversarial ranking attacks, is not surprising. Adversarial training has been widely studied and shown to enhance robustness in various downstream applications. Therefore, this finding cannot be considered as novel. If there are any novel aspects, they might lie in the quantitative experiments conducted on the improvement in retrieval tasks.

Ambiguity in the concept of curriculum learning: The paper claims to propose a CURRICULUM METRIC LEARNING method, but it is observed that there is a lack of direct connection between the pre-training task and the subsequent adversarial training task in retrieval. Additionally, the authors have not provided any contributions related to the curriculum. As a result, it is challenging to perceive the proposed method as a CURRICULUM METRIC LEARNING approach.

Insufficient analysis of experimental conclusions: Although the experimental results demonstrate performance improvement, there is a lack of in-depth analysis. The authors could further explore the reasons behind the improvement and explain why specific adversarial training initialization methods are more effective in enhancing the robustness of retrieval models. Such analysis would strengthen the credibility of the experimental conclusions.

**Questions:**

- Can the authors provide a more comprehensive analysis or theoretical justification for why adversarial training initialization is particularly crucial for enhancing robustness in retrieval tasks? This would help to better understand the underlying mechanisms and provide insights into the importance of this approach in the context of retrieval models.

-Considering the characteristics of robust feature learning and the unique property of retrieval models, are there any specific modifications or adaptations that can be made to further optimize the utilization of adversarial training for improved robustness in retrieval? It would be valuable to explore how to harness the advantages of robust feature learning and tailor it specifically for retrieval tasks, rather than applying it in a generic manner.

---

> ### Author Response · Authors · 2023-11-23
> **Response to reviewer **QZHk****
>
> We thank Reviewer **QZHk** for their insightful feedback. We are encouraged that Reviewer **QZHk** finds the problem important, our motivation clear, our method effective, and our approach accessible and easy to replicate. Below we address Reviewer **QZHk’s** concerns in detail and provide additional experimental results.
>
> **Response to Concerns:**
> * W1: We appreciate your feedback, and we revised the Contributions / Conclusions section to better highlight our contributions:
>   * We wanted to point out that our paper is the first, to the best of our knowledge, to explore and quantify the adversarial ranking robustness of adversarial trained models. This contribution is significant as it opens a new avenue in understanding how adversarial training affects learned features, and adds ranking robustness as an additional benefit to robust representations, in addition to for example better transferability [R1], or explainability [R2].
>   * We introduce a two-step approach inspired by curriculum learning, which as demonstrated experimental results significantly outperforms existing methods in defending against ranking attacks. This improvement is not merely incremental and represents a substantial improvement over baseline methods. Our approach provides a more effective (across all tested defenses and datasets) defense mechanism, which is crucial for the practical application of these models in real-world scenarios.
> * W2: Regarding your concerns about the curriculum metric learning method proposed in our paper, we recognize the need for a clearer explanation. Our approach utilizes curriculum learning by proposing to break down the learning of robust retrieval representations into solving two progressively difficult sub-tasks: (1) learning robust features using adversarial training, which is well understood and explored, (2) followed by robust retrieval training. We hypothesize adversarial training is an easier task, as we do not try to directly manipulate the feature space as one does in robust metric learning. Therefore, we hypothesize it will be easier to first learn robust features via (AT), followed by better aligning features to me semantically meaningful while remaining robust metric learning.
> * W3: We acknowledge your point on the need for a more in-depth analysis of our experimental results. In the revised version of our paper, we include a discussion on why adversarial pre-training methods are so effective in improving robust metric learning, and discussion on the proper choice of the strength of initialization robustness, and the effect of sampling strategy on learning.
>
> **Response to Questions:**
> * Q1: As outlined in response to W3, we provide a discussion on the theoretical intuition on why AT is crucial for enhancing ranking robustness.
> * Q2: This is a very good question. At this juncture, we must candidly admit that we do not have a comprehensive answer to your question.
>
>
> [R1] Salman, H., Ilyas, A., Engstrom, L., Kapoor, A., & Madry, A. (2020). Do adversarially robust imagenet models transfer better?. Advances in Neural Information Processing Systems, 33, 3533-3545.
>
> [R2] Bansal, N., Agarwal, C., & Nguyen, A. (2020). Sam: The sensitivity of attribution methods to hyperparameters. In Proceedings of the ieee/cvf conference on computer vision and pattern recognition (pp. 8673-8683).

---

### Official Review · Reviewer_fFop · 2023-10-31

**Soundness:** 3 good
**Presentation:** 3 good
**Contribution:** 3 good
**Rating:** 6
**Confidence:** 1

**Summary:**

Not familiar with the area. Please ignore my score.

**Strengths:**

Not familiar with the area. Please ignore my score.

**Weaknesses:**

Not familiar with the area. Please ignore my score.

**Questions:**

Not familiar with the area. Please ignore my score.

---

### Official Review · Reviewer_uGeY · 2023-11-07

**Soundness:** 2 fair
**Presentation:** 2 fair
**Contribution:** 2 fair
**Rating:** 5
**Confidence:** 3

**Summary:**

This paper proposes a curriculum learning method to learn robust features for image retrieval tasks. In particular, a two-stage learning scheme consisting of robust feature learning and robust metric learning is proposed to make the learned features semantic meaningful, and ranking attacks resistant. The experiments conducted on several benchmark dataset for image retrieval have verifies the effectiveness of the proposed method.

**Strengths:**

1. The idea of addressing the adversarial attack issue along with the curriculum learning direction is interesting and inspiring.
2. This paper provides the readers with a lot of experiments to demonstrate the effectiveness and the properties of the proposed method.

**Weaknesses:**

1. This paper makes the readers hard to follow due to the following reasons: a) most of the figures and tables are not clear due to a lack of description of notations. For example in Figure 2, what are EST, ACT, and HM? What does the term ERS represent? In addition, in Table 1, what are CA+, CA-, QA+, QA-, etc? b) this paper contains several typos and grammar issues. These above parts make the readers confused and easily get lost.

2. This paper needs more theoretical analysis of the proposed learning method to better support the argument of this work, not simply empirical results. Moreover, a comparison between the proposed method and some existing curriculum learning related methods could be made to better demonstrate the efficacy of this work.

3. As to the related work part, more curriculum learning related work in the robust feature learning area might be added to better locate the position of this work.

**Questions:**

Nil

**Details Of Ethics Concerns:**

Nil

---

> ### Author Response · Authors · 2023-11-23
> **Response to reviewer **uGeY****
>
> We are grateful to Reviewer **uGeY** for bringing to our notice certain issues. We are also delighted that reviewer **uGeY** finds the problem interesting, the approach effective, and the experimentation extensive. Below we include our detailed response.
>
> **Response to Concerns:**
> * We have revised Figure and Table captions to enhance the notations and descriptions thus improving paper clarity. We appreciate you pointing out the editorial issues – we revised the paper and aimed to correct the grammatical mistakes, thus making the paper easier to read.
> * As Reviewer Khkm had similar concerns, we add an additional subsection to provide intuition of why sub-task curriculum provides such a significant improvement in learning robust and meaningful representations for retrieval. We do believe a more detailed theoretical analysis is warranted and is left for future work. We however strongly believe these empirical results are interesting in their own right and the lack of theoretical analysis does in no way detract from the value of this work.
> * In order to better locate the position of our work, we add a small addition to the related work section to illustrate typical uses of curriculum learning and how our work relates to it.

---

### Official Review · Reviewer_Khkm · 2023-11-10

**Soundness:** 2 fair
**Presentation:** 2 fair
**Contribution:** 2 fair
**Rating:** 5
**Confidence:** 4

**Summary:**

The paper proposes a robust image retrieval method, which utilizes curriculum learning to design a two-stage training strategy. The designed specific training strategy aims to force the model to learn robust and semantic features simultaneously.

**Strengths:**

1. The problem is interesting and important.

2. The experiments are well-structured and prove its efficiency.

**Weaknesses:**

1. The theoretical analysis of the proposed method is lacking.

2. The key ideas of the paper are not well presented and also make the novelty not clear enough.

**Questions:**

Q1: The main concern is the lack of theoretical analysis of the proposed method in the paper. The authors have given several hypotheses and abundant empirical experimental results. However, some necessary theoretical analysis is missing.

Q2: What is the meaning of beta in Formula 2? Authors should give clear descriptions of these formulas.

Q3: In Figure 4, why does the random triplet sampling strategy achieve better performance than EST and ACT? More theoretical analysis should be given.

Q4: Please give more details about the mentioned curriculum learning strategy in the paper.

---

> ### Author Response · Authors · 2023-11-23
> **Response to reviewer **Khkm****
>
> We thank Reviewer **Khkm** for their detailed feedback and their critical assessment of the paper. We are glad that reviewer **Khkm** finds the problem important, and our experiments well-structured, and approach effective. Here we address reviewer **Khkm** concerns and questions in detail.
>
> **Response to Concerns and Questions:**
> * Weakness 1 / Question 1: As described previously, we add an additional subsection to provide intuition of why the sub-task curriculum provides such a significant improvement in learning robust and meaningful representations for retrieval. We do believe a more detailed theoretical analysis is warranted and is left for future work. We however strongly believe these empirical results are interesting in their own right and the lack of theoretical analysis does in no way detract from the value of this work.
> * Weakness 2: We revised the Abstract, Contributions, Background and Conclusions sections to better highlight the novelty of our work.
> * Question 2: Parameter “beta” in what was previously Equation 2 (Now 3 in the Appendix) is the margin parameter in the triplet loss.
> * Question 3: We hypothesize the Random sampling strategy outperforms the Soft-hard sampling strategy due to the conflict of how Soft-hard sampling generates triplets and the goal of the adversary in EST / ACT. Soft-hard sampling generates triplets that “likely” have hard positives and hard negatives [R1], that are likely hard to collapse (ACT), making the random sampling more effective in this case. Please see Appendix A1 for further discussion.
> * Question4: Please see the Background section for the expanded description of the curriculum learning strategy.
>
> [R1] Roth, K., Milbich, T., Sinha, S., Gupta, P., Ommer, B., & Cohen, J. P. (2020, November). Revisiting training strategies and generalization performance in deep metric learning. In International Conference on Machine Learning (pp. 8242-8252). PMLR.

---

> > ### Comment · Reviewer_Khkm · 2023-11-23
> > **Response to the Rebuttal**
> >
> > Dear Authors,
> >
> > Thank you for providing a rebuttal response and it addresses some of the concerns. I also read the other reviews there. It's highly recommended to conduct further exploration and give a more detailed theoretical analysis to improve the work.
> >
> > I will consider to keep my current score.

---

### Author Response · Authors · 2023-11-23
**We are grateful for the reviews!**

We thank the reviewers for their valuable feedback and constructive criticism. We are thrilled that the reviews find the problem important (**Khkm, uGeY, QZHk**), our experiments thorough and supportive of the approaches’ efficiency (**Khkm, uGeY, and QZHk**), and the approach easy to replicate making it accessible to a wider audience (**QZHk**).

We have carefully considered each comment and we made revisions to our paper to address the concerns raised. By incorporating these changes, we believe that our paper will be significantly improved in terms of its analysis, presentation, clarity, novelty, and overall contribution to the field. We appreciate the opportunity to revise our work and are confident that these enhancements will address the reviewers' concerns effectively.

**Addressing the Novelty of Our Work:**

Reviewer QZHk pointed out that the use of adversarially trained models to enhance ranking robustness is not surprising. However, our work introduces a unique two-stage learning procedure based on curriculum learning to this underexplored domain (robust defenses against retrieval attacks). We are the first, to the best of our knowledge, to specifically address and quantify the adversarial ranking robustness of adversarially trained models. This contribution is significant as it opens a new avenue in understanding how adversarial training affects learned features, and adds ranking robustness as an additional benefit of robust representations, in addition to for example better transferability [R1], or explainability [R2].

Additionally, our experimental results demonstrate that our approach significantly outperforms existing methods in defending against ranking attacks. This improvement is not merely incremental and represents a substantial improvement over baseline methods. Our approach provides a more effective (across all tested defenses and datasets) defense mechanism, which is crucial for the practical application of these models in real-world scenarios.

**Addressing Theoretical Analysis:**

Although there is not a detailed theoretical analysis, we believe that does not detract from the value of our paper, as often empirical results precede theoretical understanding. We believe that the empirical results are interesting in their own right, and wanted to point out that all of the reviewers agree that our experiments are supportive of the approach’s efficacy.

**Paper updates:**

**[Abstract, Contributions, and Conclusions]** We revise the Abstract, Contributions subsection, and Conclusions section to better outline our contributions:
* We are the first, to the best of our knowledge, to explore and quantify the adversarial ranking robustness of robust features.
* We propose a two-stage learning procedure based on curriculum learning, which decomposes the task of simultaneously learning robust and meaningful representations into two sub-tasks: (1) learning robust features using adversarial training, as it is well explored and shown to scale to large datasets, and (2) learning meaningful features that while preserving robustness.

**[Updated Figure and Table captions]** Responding to Reviewer 2's concerns, we made our best effort  to revise all figure and table descriptions for clarity. We added detailed descriptions of the notations used in Figures and Tables. For instance, EST, ACT, HM, and ERS are now clearly defined in the figure captions and the text. This revision aims to enhance the readability and understanding of our experimental setup and results.

**[Updated Figure 4.]** We updated Figure 4 with an updated layout for better space management.

**[Updated Background section]** Background section now contains a subsection which provides a better explanation of curriculum learning (based on a sub-task curriculum) and how adversarial pre-training (sub-task 1) influences robust metric learning (sub-task 2).

**[Updated Experimental section (Section 4)]** We add two subsections that: (a) provide theoretical intuition of why robust pre-training (initialization) (sub-task 1) improves robust metric learning (sub-task 2), and (b) what is the best choice of robustness parameter for robust pre-training. We also add an additional section to the Appendix where we discuss the effect on sampling strategy on learning.

[R1] Salman, H., Ilyas, A., Engstrom, L., Kapoor, A., & Madry, A. (2020). Do adversarially robust imagenet models transfer better?. Advances in Neural Information Processing Systems, 33, 3533-3545.

[R2] Bansal, N., Agarwal, C., & Nguyen, A. (2020). Sam: The sensitivity of attribution methods to hyperparameters. In Proceedings of the ieee/cvf conference on computer vision and pattern recognition (pp. 8673-8683).

---

### Meta-Review · Area_Chair_VkME · 2023-12-17

**Metareview:**

The paper investigates robustness of deep metric learning models to adversarial ranking attacks and proposes a two-stage curriculum-like training to address this problem.

While the reviewers acknowledged the importance to study robust features for image retrieval tasks, they raised several concerns that were viewed by AC as the critical issues:
(1) lack of technical contribution – neither theoretical justification (all reviewers) nor sufficient novelty (see detailed comment by the reviewer QZHk) has been demonstrated;
(2) clarity and rigor in presentation – see Reviewer Khkm and Reviewer uGeY comments;
(3) insufficient analysis of experimental conclusions to assess the efficacy of the proposed approach – see Reviewer QZHk suggestions how to improve, and Reviewer uGeY suggestion to provide a comparison to existing curriculum learning methods.
The (only positive) review provided by Reviewer fFop was not taken into consideration based on lack of expertise in this area and the Reviewer requests.

In conclusion, the reviewers unanimously question the novelty and the importance of the method. The rebuttal was able to clarify some questions, but did not manage to sway any of the reviewers. A general consensus among reviewers and AC suggests, in its current state the manuscript is not ready for a publication. We hope the reviews are useful for improving and revising the paper.

**Justification For Why Not Higher Score:**

The reviewers unanimously question the novelty and the importance of the method. The rebuttal was able to clarify some questions, but did not manage to sway any of the reviewers.

**Justification For Why Not Lower Score:**

N/A

---

### Decision · Program_Chairs · 2024-01-16

Reject